# Propranolol, chlorpromazine and diclofenac restore susceptibility of extensively drug-resistant (XDR)-*Acinetobacter baumannii* to fluoroquinolones

**Mostafa A. Mohammed**[1], **Mohammed T. Ahmed**[1], **Bahaa E. Anwer**[1], **Khaled M. Aboshanab**[2]*, **Mohammad M. Aboulwafa**[2]

1 Department of Microbiology and Immunology, Faculty of Pharmacy, Al Azhar University, Assiut, Egypt,
2 Department of Microbiology and Immunology, Faculty of Pharmacy, Ain Shams University, Cairo, Egypt

* aboshanab2012@pharma.asu.edu.eg

**Data Availability Statement:** All relevant data are within the manuscript

## Abstract

Nosocomial infections caused by extensively drug-resistant (XDR) or Pan-Drug resistant (PDR) *Acinetobacter* (*A.*) *baumannii* have recently increased dramatically creating a medical challenge as therapeutic options became very limited. The aim of our study was to investigate the antibiotic-resistance profiles and evaluate the various combinations of ciprofloxacin (CIP) or levofloxacin (LEV) with antimicrobial agents and non-antimicrobial agents to combat antimicrobial resistance of XDR *A. baumannii*. A total of 100 (6.25%) *A. baumannii* clinical isolates were recovered from 1600 clinical specimens collected from hospitalized patients of two major university hospitals in Upper Egypt. Antimicrobial susceptibility tests were carried out according to CLSI guidelines. Antimicrobial susceptibility testing of the respective isolates showed a high percentage of bacterial resistance to 19 antimicrobial agents ranging from 76 to99%. However, a lower percentage of resistance was observed for only colistin (5%) and doxycycline (57%). The isolates were categorized as PDR (2; 2%), XDR (68; 68%), and multi-drug resistant (MDR) (30; 30%). Genotypic analysis using ERIC-PCR on 2 PDR and 32 selected XDR isolates showed that they were not clonal. Combinations of CIP or LEV with antibiotics (including, ampicillin, ceftriaxone, amikacin, or doxycycline) were tested on these *A. baumannii* non-clonal isolates using standard protocols where fractional inhibitory concentrations (∑FICs) were calculated. Results of the respective combinations showed synergism in 23.5%, 17.65%, 32.35%, 17.65% and 26.47%, 8.28%, 14.71%, 26.47%, of the tested isolates, respectively. CIP or LEV combinations with either chlorpromazine (CPZ) 200 μg/ml, propranolol (PR) in two concentrations, 0.5 mg/ml and 1.0 mg/ml or diclofenac (DIC) 4 mg/ml were carried out and the MIC decrease factor (MDF) of each isolate was calculated and results showed synergism in 44%, 50%, 100%, 100% and 94%, 85%, 100%, 100%, of the tested isolates, respectively. In conclusion, combinations of CIP or LEV with CPZ, PR, or DIC showed synergism in most of the selected PDR and XDR *A. baumannii* clinical isolates. However, these combinations have to be re-evaluated in vivo using appropriate animal models infected by XDR- or PDR- *A. baumannii*.

**Funding:** The author(s) received no specific funding for this work.

**Competing interests:** The authors have declared that no competing interests exist.

## Introduction

*A. baumannii* is a Gram-negative, strictly aerobic bacterial pathogen that can survive for prolonged periods under a wide range of environmental conditions and on surfaces making it a frequent cause of nosocomial infections and outbreaks [1]. *A. baumannii* is one of the most common bacteria causing hospital acquired and difficult to treat infections, due to its endless capacity to acquire antimicrobial resistance owing to the plasticity of its genome [1]. Many acquired resistance mechanisms have been reported for this pathogen, rendering it able to express MDR or XDR phenotypes which are associated with significant morbidities and mortalities [2].

In the last four decades, fluoroquinolones (FQs) have shown good activity against *A. baumannii* isolates, even better than penicillin derivatives, cephalosporins, and aminoglycosides. However, resistance to these drugs has rapidly emerged in recent years [3]. The developing resistance of *A. baumannii* to polymyxins has been described at least during the last two decades and this was attributed, besides their misuse, to the abundance of these antibiotics in multiple pharmaceutical markets [4] Consequently, researchers have been focusing their work on finding new therapeutic options to overcome this health problem [5–7]. However, little information is available on the treatment regimens of *A. baumannii* by FQs combination with antimicrobial or non-antimicrobial agents. For example, the combination of LEV with amikacin in north Egypt to control *A. baumannii* resistance was barely successful [8], while several other attempts failed [9]. However, a fully successful combination of CIP with amikacin against Gram-negative bacteria in Upper Egypt was reported [10]. On the other hand, another study reported the successful use of polymyxin B and doxycycline combination in patients with MDR *A. baumannii* infections [11]. Recently, various reports confirmed that PR, a non-selective beta-blocker, had potent negative effects on the cell growth viability and progression, and was suggested with evidence to reduce cancer types [12, 13]. Some studies were conducted on the use of a combination of a non-selective beta-blocker, including PR [7] or carvedilol [14], with other drugs for the treatment of rosacea. Most patients with bacterial infections suffer pain and fever that require complex treatment with antibiotics, antipyretics, and analgesics. Antipyretics and non-steroidal anti-inflammatory drugs (NSAIDs) commonly co-administered with antimicrobial therapy often modify the susceptibility of microbes to antimicrobial therapy [15–17] by changing the hydrophobicity of microbes [18], influencing biofilm production [19], and interacting with the transport and release of antibiotics [20]. The antibacterial activity of antipsychotic agents, such as CPZ has been recently evaluated on certain Gram-negative pathogens [5, 6], Gram-positive pathogens and *Mycobacterium tuberculosis* [21]. However, the effect of FQs in combination with non-antimicrobial agents such as PR, DIC, or CPZ on *A. baumannii* clinical isolates particularly, XDR or PDR phenotypes is still not yet explored.

Therefore, this study aimed to determine the antibiotic-resistant phenotypes of *A. baumannii* recovered from hospitalized patients of two major University Hospitals in Upper Egypt, followed by evaluating various CIP-LEV combinations with several antibiotics and non-antibiotics particularly, PR, DIC, and CPZ as an attempt to control the resistance of this pathogen.

## Methods

### Bacterial isolates

One thousand and six hundred clinical specimens including blood, ventral venous catheters, endotracheal tube, urinary tract catheter, sputum, urine, skin, wound and throat swabs were collected from hospitalized febrile patients (oral temperature >38˚C for at least 1 hour) between January 2014 and March 2019 from Al Azhar and Assiut University Hospitals, Upper Egypt. The study was approved by the Research Ethics Committee Faculty of Pharmacy, Ain

Shams University ENREC-ASU-63 where both informed and written consents were obtained from parents of patients after explaining the study purpose. All specimens were streak-plated on MacConky agar (Oxoid Limited, England), then the non-lactose fermenter colonies sub-cultured on blood agar and finally on Herellea agar (Himedia, India), and incubated at 37˚C for 24 hours [22].

## Phenotypic and genotypic identification of *A. baumannii*

For phenotypic identification, separate colonies were processed for qualitative conventional diagnostic tests for *A. baumannii*; including Gram staining and biochemical tests such as catalase, citrate utilization, oxidase, and indole tests [22]. For genotypic identification, the amplification of the intrinsic $bla_{OXA-51}$-like gene was used [23]. The primers used for detection of $bla_{OXA-51}$ gene were F-(5-TAATGCTTTGATCGGCCTTG-3) and R-(5-TGGATTGCACTTC ATCTTGG-3) [24]. DNA was extracted and purified using the GeneJet PCR purification kit (Thermo, USA, catalog No. K0721), following instructions of the manufacturer. The extracted DNA was stored at -20˚C for further use. PCR amplification was performed in the thermal cycler (Nyx Technik ATC 401, USA) using 25 pmol of each primer, 100 ng of genomic DNA, the PCR mixture (25 μl) formulated according to the protocol supplied with the Dream *Taq* master mix kit (Thermo Fisher, UK) and 1 μl template DNA. PCR conditions were 94˚C for 3 min, 35 cycles at 94˚C for 45 s at 57˚C for 45 s, and at 72˚C for 1 min, followed by a final extension step at 72˚C for 5 min. PCR products were analyzed using agarose gel electrophoresis [23]. The characteristic band at 353 bp for $bla_{OXA-51}$-like genes was considered positive only for *A. baumannii*. *A. baumannii* ATCC 19606 was used for quality control.

## Antimicrobial susceptibility testing

**Disc diffusion.** This is was carried out using Kirby-Bauer disk diffusion method as recommended by the Clinical and Laboratory Standards Institute (CLSI) [25]. The antibiotic discs used for susceptibility testing were imipenem (10 μg), meropenem (10 μg), piperacillin (100 μg), piperacillin/tazobactam (100/10 μg), ampicillin/sulbactam (10/10 μg), ceftazidime (30 μg), cefotaxime (30 μg), ceftriaxone (30 μg), cefepime (30μg), amikacin (10μg), tobramycin (10 μg), gentamicin (10 mg), CIP (5 μg), LEV (5 μg), gatifloxacin (5 μg), trimethoprim/sulfamethoxazole (12.5/23.75 μg), tigecycline (15 μg), doxycycline (30 mcg) and colistin (10 units). All antimicrobial discs were purchased from Oxoid (UK) except gatifloxacin discs which were purchased from Himedia (India). MDR, XDR, and PDR phenotypes were identified as previously determined [26].

**Minimum inhibitory concentrations (MICs).** MICs of CIP, LEV, ampicillin (AMP), ceftriaxone (CRO), amikacin (AK), doxycycline (DO), and vancomycin (VC) against some selected XDR- and PDR- *A. baumannii* isolates were determined by broth micro-dilution method according to CLSI guidelines [27]. *A. baumannii* ATCC 19606 was used for quality control. The MIC for antimicrobial agents ranged from 0.125–256 μg/ml.

## Molecular typing of recovered isolates

To investigate the clonal relationship, clonal expansion, and diversity of the recovered *A. baumannii* isolates, molecular typing using ERIC-PCR was carried out on some of these isolates that showed PDR and XDR phenotypes [28]. Genomic DNA was extracted using the Genomic DNA Purification Kit (Thermo Fisher Scientific, UK) according to the manufacturer's instructions. ERIC-PCR was carried out using the ERIC-1 (5′-ATGTAAGCTCCTGGGGATTCAC-3′) and ERIC-2 (5′-AAGTAAGTG ACTGGGGTGAGCG-3′) primers as previously described [28]. The PCR products were analyzed using agarose gel electrophoresis using 1.5% (w/v) agarose

containing ethidium bromide (0.5 mg/ml) and visualized on a UV transilluminator. Analysis of ERIC-PCR dendrogram was constructed by the use of the UPGMA clustering method, Bionumeric program version 7.6 (Applied Maths). The Percentage of similarity among 34 strains of *A. baumannii* was calculated by the use of Jaccard's Coefficient [29].

## Evaluation of drug combinations

**Evaluation of FQs-antibiotic combinations.**    Firstly, the MIC of each antimicrobial agent CIP, LEV, AMP, CRO, VC, AK, and DO was determined using the broth microdilution technique according to the CLSI guidelines 2011 [27]. In vitro, combinations of FQs members including either CIP or LEV with either AMP, CRO, VC, AK, or DO were evaluated through the checkerboard method and the protocol described by Hsieh et al [30]. The resulting checkerboard included each combination of two antibiotics in opposite directions where one antibiotic was serially diluted horizontally and the other was diluted vertically. The sum of the fractional inhibitory concentration (ΣFICs) was calculated according to the following equation:

$$\sum \text{FICs} = \text{FIC}_A + \text{FIC}_B = \frac{A}{MICA} + \frac{B}{MICB}$$

$\text{FIC}_A$ is the MIC of drug A in the combination /MIC of drug A alone, and $\text{FIC}_B$ is the MIC of drug B in the combination /MIC of drug B alone. The combination is considered synergistic when ΣFIC is ≤ 0.5, additive when ΣFIC is > 0.5 and ≤ 1, indifferent when ΣFIC is >1 and ≤ 4, and antagonistic when ΣFIC is > 4 [9].

**Evaluation of FQs- non-antibiotic combinations.**    FQs combinations with different non-antibiotic such as the non-selective β blocker PR, 25 μg-1 mg/ml (Sigma, Aldrich, UK), selective β blocker labetalol, 25 μg-1 mg/ml (Sigma, Aldrich, UK), the NSAIDs DIC, 25 μg-4 mg/ml (Novartis, Egypt), acetylsalicylic acid, 100–400 μg/ml (Adwia, Egypt), the proton pump inhibitor omeprazole and esomeprazole 25–200 μg/ml (Sigma, Aldrich, UK), the diuretic furosemide, 25 μg-1 mg/ml (Sanofi Aventis, Egypt) and the antipsychotic CPZ, 25–200 μg/ml (Sigma, Aldrich, UK) against some selected *A. baumannii* were evaluated by calculating the MDF [31]. The MDF of each isolate was calculated according to the following formula MDF = $\text{MIC}_{\text{without non-antibiotic}}$ / $\text{MIC}_{\text{with non-antibiotic}}$. An MDF value greater than 4 was defined as a significant inhibition according to the protocol of Huguet [31]. The MIC of tested FQs ranged from 0.03–64 μg/ml.

## Results

### Specimen collection and identification of the recovered *A. baumannii* isolates

The endotracheal tube specimen showed the highest percentage (29%) of the recovered *A. baumannii* isolates while throat swab, skin, and central venous catheter specimens gave the lowest percentages (2.0% each) (Fig 1). Out of 1600 samples cultivated on MacConky agar, only 623 isolates (38.94%) were identified as non-lactose fermenters, of which 151 isolates were phenotypically identified as *Acinetobacter* isolates. Of these, only 100 isolates (6.25%) were PCR positive for the $bla_{\text{OXA-51}}$-like gene and therefore were identified as *A. baumannii* (Fig 2).

### Antimicrobial susceptibility testing

**Disc diffusion.**    As shown in Fig 3, all the *A. baumannii* isolates exhibited high resistance to most of the tested antimicrobial agents (76–99%). However, the lowest resistance recorded was to colistin (5%) followed by doxycycline (57%). Phenotypically, 30 (30%), 68(68%), and 2

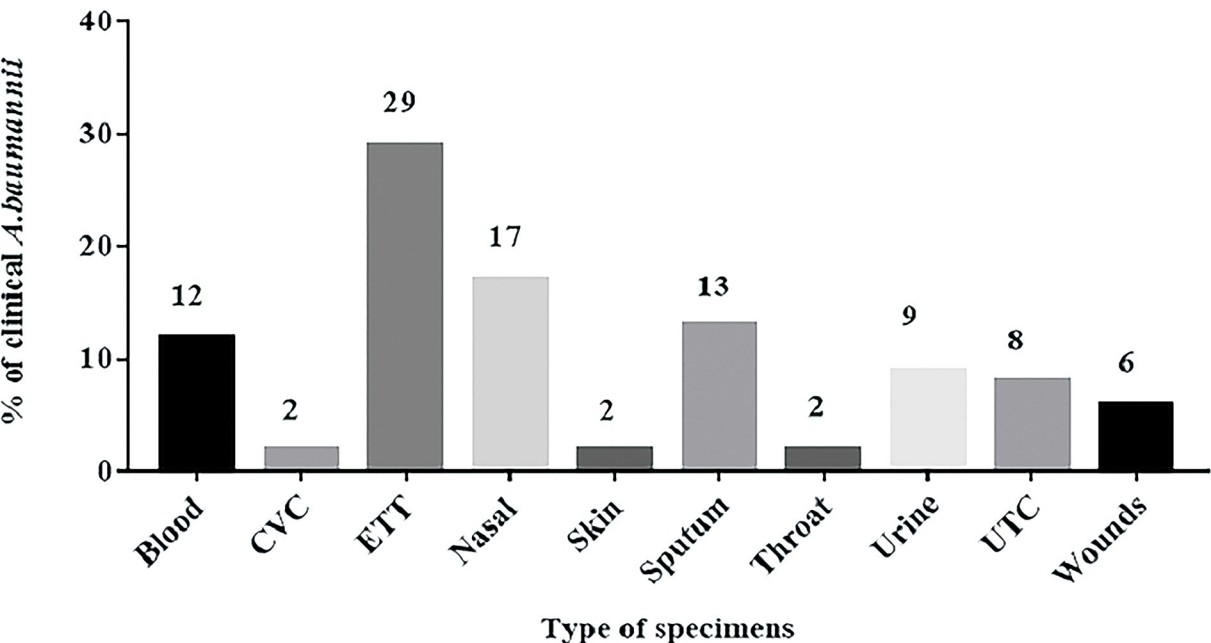

**Fig 1. Frequencies of *A. baumannii* isolates from patients with different clinical samples.** CVC: Central Venus Catheters. ETT: Endotracheal tube, UTC: Urinary Tract Catheter.

(2%) isolates were reported as MDR, XDR, and PDR, respectively (Fig 4). Analysis of the resulting susceptibility to 19 antimicrobial agents showed the resistance diversity of these isolates. They were divided into 12 major profiles, according to the number of antimicrobial agents they exhibited resistance to, ranging from 19 to 6 antimicrobial agents. The first profile represented PDR isolates (two isolates) that were resistant to all 19 antimicrobial agents tested. The second profile represented some of XDR isolates (32 isolates) that were resistant to all tested antimicrobial agents except colistin. Finally, profile number 12 was resistant to only 6 out of the 19 tested antimicrobial agents (Fig 5).

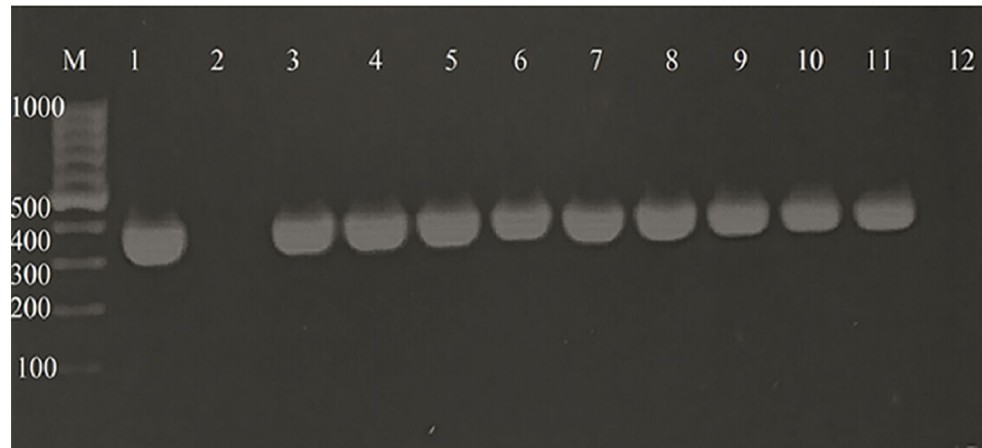

**Fig 2. PCR amplification of *bla*$_{OXA-51}$ like gene in some *A. baumannii* clinical isolates, lane M, a gene ruler 100 bp ladder; lane 1, a positive control; lane 2, a negative control; lanes 3 to 11, positive results with an expected size of 353 bp; lane 12, negative results.**

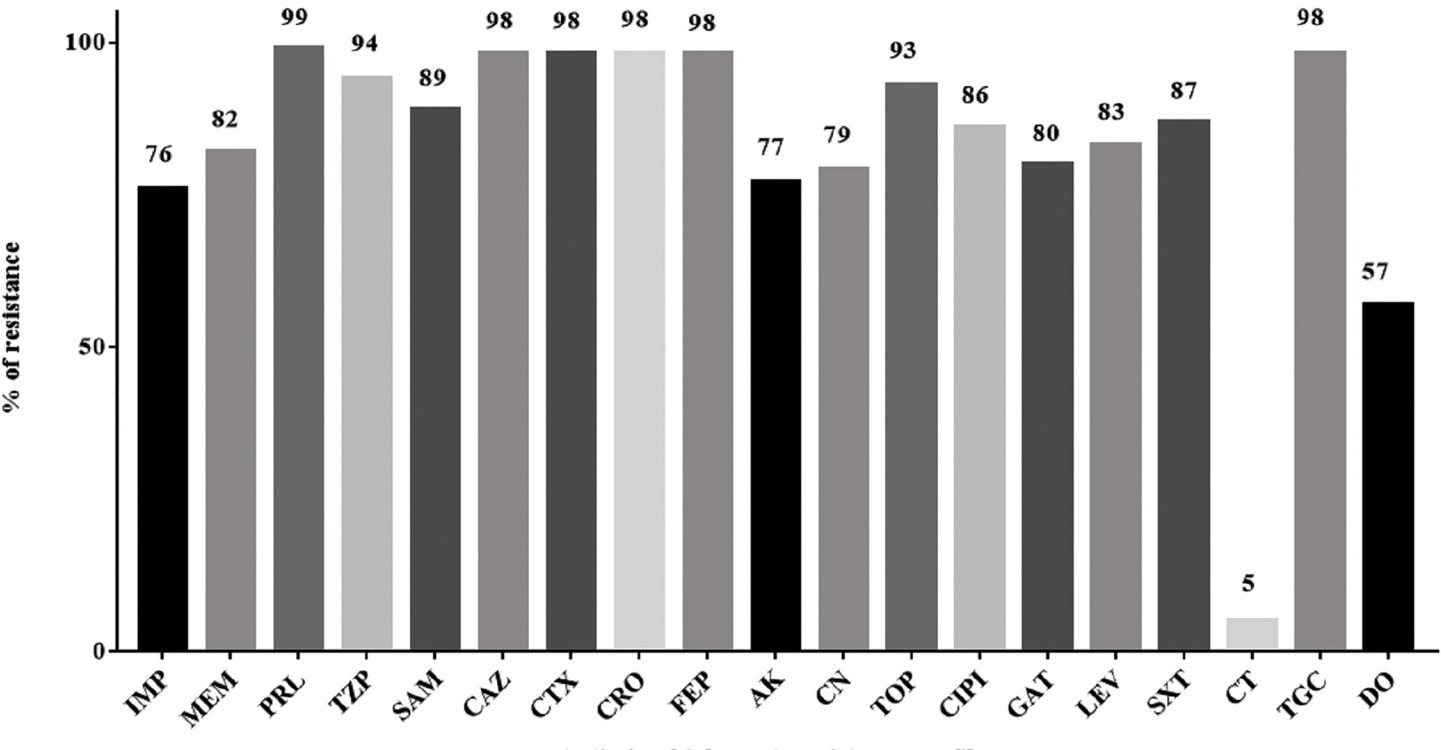

**Fig 3. Antimicrobial susceptibility testing of *A. baumannii* isolates (n = 100).** IMP: Imipenem, MEM: Meropenem, PRL: Piperacillin, TZP: Tazopactam/Piperacillin, SAM: Sulbactam/Ampicillin, CAZ: Ceftazidime, CTX; Cefotaxime, CRO: Ceftriaxone, FEP: Cefepime, AK; Amikacin, CN: Gentamicin, TOB: Tobramycin, CIP: Ciprofloxacin, GAT: Gatifloxacin, LEV: Levofloxacin, SXT: Sulfamethoxazole/Trimethoprim, CT: Colistin, TGC: Tigecycline, DO: Doxycycline.

**Minimum inhibitory concentration.** The MIC values of 34 *A. baumannii* isolates including 2 PDR (profile 1, Fig 5) and 32 XDR isolates (Profile 2; Fig 5) are outlined in Table 1. The MIC values of LEV, CIP, AMP, CRO, AK, DO and CV against the selected 34 isolates ranged from 4–32, 8–64, 32–128, 64–256, 32–256, 16–256, and 16–256 μg/ml, respectively.

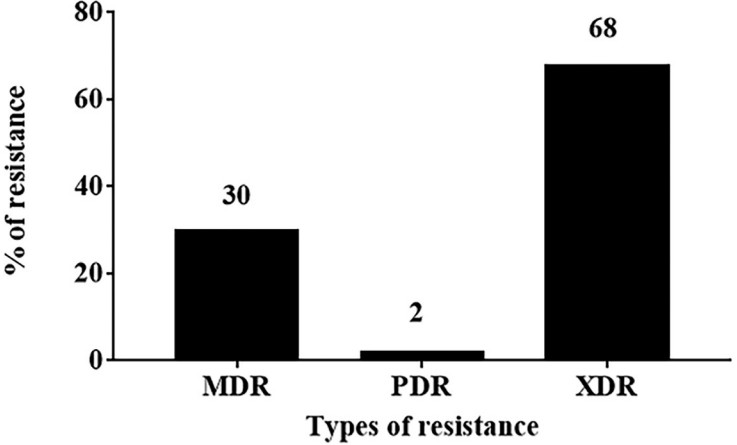

**Fig 4. Phenotypic analysis of the recovered *A. baumannii* isolates (n = 100).** MDR (Multi-Drug resistant), XDR; extensively–Drug-resistant and PDR (Pan-Drug resistant).

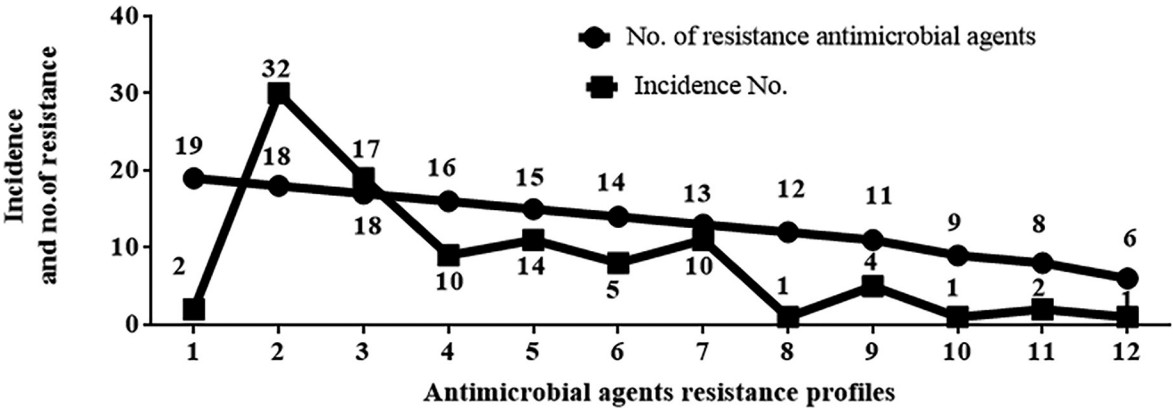

**Fig 5. Antimicrobial agents resistance profiles scored for different *A. baumannii* isolates.**

### Genotyping of isolates

A genotypic analysis of the 34 selected isolates using ERIC-PCR is shown in Fig 6. The obtained isolates were divided into three major clusters (Cluster I, II, and III) with similarity ranging from 0% to 97.3%. Cluster I was a major cluster representing 58.82% (20/34) of the isolates. Cluster II represented 5.88% (2/34) while cluster III represented 35.29% of the total isolates. Clusters I and II were completely recovered from Assiut University hospitals. Cluster III was recovered from Al Azhar University hospital except for one isolate (AS-47) which was recovered from Assiut University hospital.

### Evaluation of drug combinations

**Evaluation of FQs-antibiotic combinations.** The results of the antimicrobial combination with FQs (LEV, CIP) using the checkerboard method were presented in the Tables 2 and 3.

Analysis of Tables 1–3 were carried out and the MICs of FQs after the addition of antimicrobial agents were shown in Table 4.

**Evaluation of FQs- non-antibiotic combinations.** The results of FQs in combination with non-antibiotics are presented in Tables 5 and 6. Results revealed that, CPZ 200 μg/ml, when combined with CIP or LEV, increased the susceptibilities of the isolates to the antimicrobials by 44.12% and 94.12%, respectively. No effect was obtained by using concentrations ranging from 25–150 μg/ml. PR, in a concentration of 0.5 mg /ml, diminished the resistance by 50% and 85.29% when combined with CIP and LEV, respectively (Table 7). Interestingly, PR (1 mg/ml) and sodium DIC (4 mg/ml) completely diminished FQ resistance when each was used in combination with either CIP or LEV (Tables 5 and 6). No significant effects were observed when each of omeprazole, esomeprazole, acetylsalicylic acid, furosemide, and labetalol were used in the combination with CIP or LEV.

### Discussion

Respiratory tract, urinary tract, and blood infections are the most frequent clinical complications of *A. baumannii*. In our study, a total of 100 *A. baumannii* MDR clinical isolates were recovered from respiratory tract specimens (including endotracheal tubes, nasal, sputum and throat), urinary tract specimens (including urine and urinary tract catheter), blood, wound, skin, and central venous catheter specimens. Previous studies reported that, the respiratory tract, blood, and urine specimens were the main sources of *A. baumannii* pathogens [32, 33]. Moreover, wounds or soft-tissue infections, skin and catheter-associated infections (including,

**Table 1. MIC of selected antimicrobial agents against PDR and some selected XDR isolates (n = 34).**

| No. | Isolates code | MICs (µg/ml) | | | | | | |
|---|---|---|---|---|---|---|---|---|
| | | CIP | AMP | LEV | CRO | AK | DO | VC |
| 1 | AS-07 | 16 | 64 | 8 | 64 | 256 | 64 | 128 |
| 2 | AS-09 | 32 | 64 | 8 | 128 | 256 | 256 | 128 |
| 3 | AS-15 | 16 | 64 | 8 | 64 | 256 | 256 | 128 |
| 4 | AS-18 | 32 | 64 | 16 | 64 | 256 | 256 | 128 |
| 5 | AS-19 | 32 | 64 | 16 | 256 | 256 | 256 | 128 |
| 6 | AS-24 | 32 | 64 | 16 | 64 | 32 | 16 | 16 |
| 7 | AS-25 | 16 | 64 | 16 | 64 | 64 | 16 | 16 |
| 8 | AS-26 | 32 | 64 | 8 | 64 | 256 | 256 | 16 |
| 9 | AS-30 | 32 | 64 | 16 | 64 | 256 | 256 | 16 |
| 10 | AS-31 | 8 | 64 | 4 | 128 | 64 | 256 | 16 |
| 11 | AS-32 | 64 | 128 | 32 | 256 | 256 | 256 | 64 |
| 12 | AS-34 | 16 | 64 | 4 | 64 | 64 | 256 | 16 |
| 13 | AS-35 | 16 | 64 | 4 | 64 | 256 | 256 | 16 |
| 14 | AS-36 | 16 | 64 | 4 | 64 | 256 | 256 | 16 |
| 15 | AS-37 | 16 | 64 | 4 | 64 | 256 | 64 | 16 |
| 16 | AS-38 | 16 | 32 | 4 | 64 | 128 | 64 | 64 |
| 17 | AS-39 | 16 | 32 | 4 | 64 | 128 | 64 | 128 |
| 18 | AS-42 | 32 | 128 | 8 | 64 | 256 | 64 | 256 |
| 19 | AS-47 | 32 | 128 | 8 | 64 | 256 | 64 | 256 |
| 20 | AS-50 | 16 | 128 | 8 | 64 | 256 | 64 | 256 |
| 21 | AS-51 | 16 | 128 | 8 | 64 | 128 | 16 | 256 |
| 22 | AS-52 | 32 | 128 | 16 | 64 | 128 | 16 | 256 |
| 23 | AS-54 | 16 | 128 | 8 | 64 | 128 | 16 | 256 |
| 24 | AZ-02 | 16 | 32 | 8 | 256 | 64 | 16 | 32 |
| 25 | AZ-06 | 32 | 32 | 8 | 256 | 64 | 16 | 256 |
| 26 | AZ-10 | 16 | 32 | 8 | 64 | 256 | 16 | 128 |
| 27 | AZ-25 | 32 | 64 | 4 | 64 | 128 | 32 | 256 |
| 28 | AZ-26 | 16 | 32 | 8 | 64 | 256 | 32 | 256 |
| 29 | AZ-36 | 16 | 32 | 8 | 64 | 256 | 16 | 128 |
| 30 | AZ-41 | 32 | 32 | 4 | 64 | 256 | 16 | 64 |
| 31 | AZ-42 | 32 | 32 | 8 | 64 | 256 | 16 | 128 |
| 32 | AZ-43 | 32 | 32 | 4 | 64 | 256 | 64 | 16 |
| 33 | AZ-44 | 32 | 64 | 4 | 64 | 128 | 64 | 16 |
| 34 | AZ-46 | 16 | 32 | 16 | 64 | 128 | 64 | 16 |

MICs, minimum inhibitory concentrations; CIP, ciprofloxacin; LEV, levofloxacin; AMP, ampicillin; CRO, ceftriaxone; AK, amikacin; DO, doxycycline; and VC, vancomycin.

a central venous catheter, and urinary tract catheter) were recently reported to be caused by *A. baumannii* and therefore, highlighted the importance of environmental contamination in disseminating such infections. Another study revealed that, the epidemiology of *A. baumannii* infection differs according to the anatomical site as well as the clinical conditions of the patients worldwide [34]. The *A. baumannii* pathogen was recorded as an opportunistic pathogen often susceptible to colistin and having a low susceptibility to other antimicrobial agents [35, 36]. Therefore, *A. baumannii's* growing resistance is a worldwide problem [37]. The incurable strains of *A. baumannii* endanger the lives of millions of hospitalized patients every year

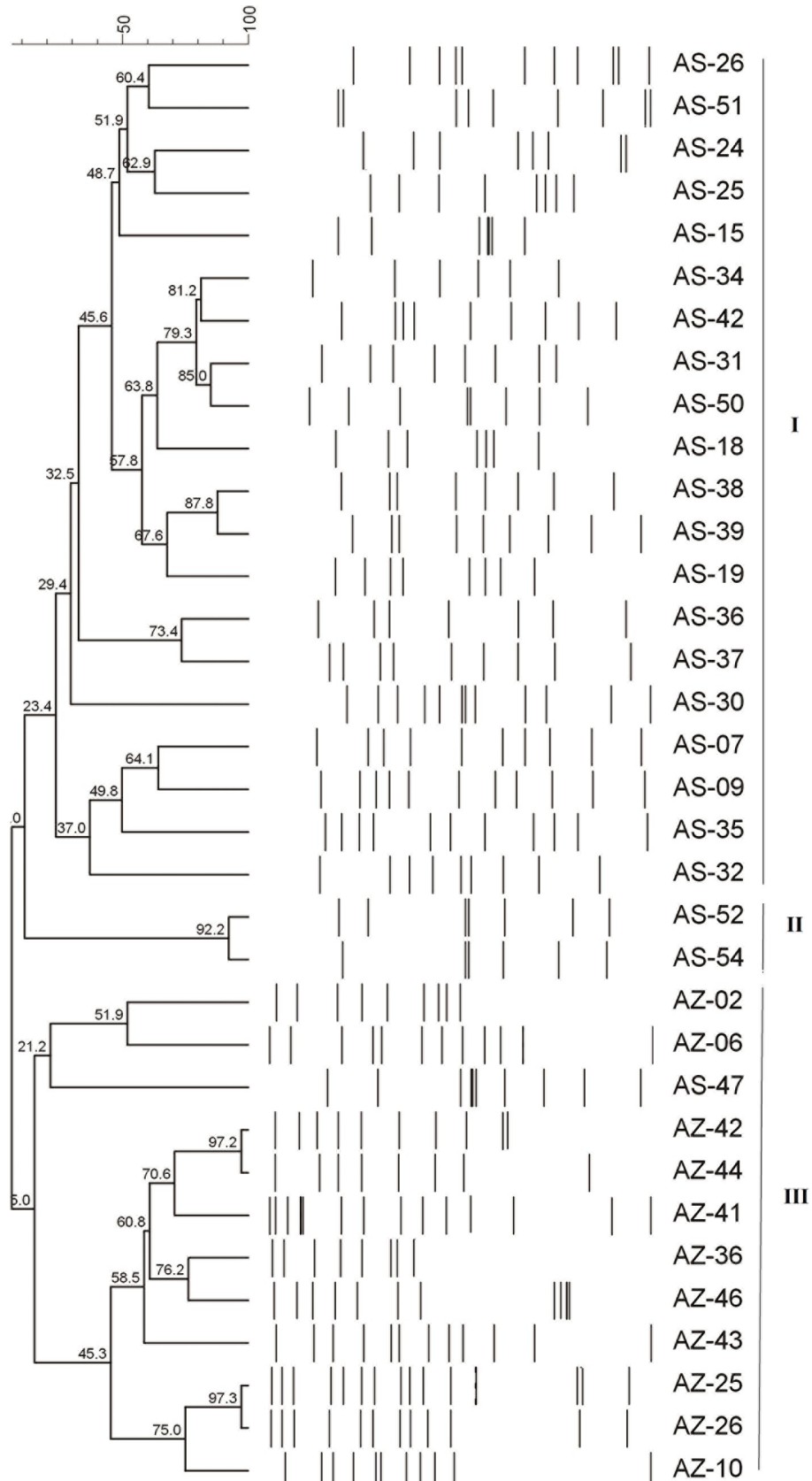

**Fig 6. ERIC-PCR dendrogram analysis for 34 *A. baumanni* was constructed by the UPGMA clustering method.** Percent of similarity among the 34 isolates of *A. baumannii* were calculated by using Jaccard's Coefficient using Bionumeric program software.

[38]. Continuous study of their resistance pattern is a must to control or at least decrease their devastating effect on the quality of medical treatment. In the current study, the antimicrobial

**Table 2. Effects of CIP combinations with different antimicrobial agents by the checkerboard method.**

| No. | Isolates code | CIP-AMP μg/ml | ΣFIC | Int | CIP-CRO μg/ml | ΣFIC | Int | CIP-AK μg/ml | ΣFIC | Int | CIP-DO μg/ml | ΣFIC | Int | CIP-VC μg/ml | ΣFIC | Int |
|---|---|---|---|---|---|---|---|---|---|---|---|---|---|---|---|---|
| 1 | AS-07 | 8(64) | 1.50 | I | 8(64) | 1.50 | I | 4(64) | 0.50 | S | 4(32) | 0.75 | D | 16(128) | 2.00 | I |
| 2 | AS-09 | 16(64) | 1.50 | I | 16(64) | 1.00 | D | 8(64) | 0.50 | S | 16(64) | 0.75 | D | 16(128) | 1.50 | I |
| 3 | AS-15 | 8(64) | 1.50 | I | 16(64) | 2.00 | I | 8(128) | 1.00 | D | 8(32) | 0.63 | D | 32(128) | 3.00 | I |
| 4 | AS-18 | 8(32) | 0.75 | D | 16(64) | 1.50 | I | 32(64) | 1.25 | I | 32(64) | 1.25 | I | 16(128) | 1.50 | I |
| 5 | AS-19 | 16(32) | 1.00 | D | 16(128) | 1.00 | D | 8(64) | 0.50 | S | 16(64) | 0.75 | D | 16(128) | 1.50 | I |
| 6 | AS-24 | 8(16) | 0.50 | S | 16(64) | 1.50 | I | 16(16) | 1.00 | D | 4(8) | 0.63 | D | 16(16) | 1.50 | I |
| 7 | AS-25 | 4(16) | 0.50 | S | 4(64) | 1.25 | I | 32(32) | 2.50 | I | 2(4) | 0.38 | S | 16(16) | 2.00 | I |
| 8 | AS-26 | 8(16) | 0.50 | S | 4(16) | 0.38 | S | 8(64) | 0.50 | S | 16(64) | 0.75 | D | 8(8) | 0.75 | D |
| 9 | AS-30 | 4(16) | 0.38 | S | 4(16) | 0.38 | S | 8(256) | 1.25 | I | 16(64) | 0.75 | D | 8(8) | 0.75 | D |
| 10 | AS-31 | 2(16) | 0.50 | S | 4(16) | 0.63 | D | 2(16) | 0.50 | S | 16(64) | 2.25 | I | 2(16) | 1.25 | I |
| 11 | AS-32 | 32(64) | 1.00 | D | 16(128) | 0.75 | D | 16(128) | 0.75 | D | 32(64) | 0.75 | D | 32(32) | 1.00 | D |
| 12 | AS-34 | 8(32) | 1.00 | D | 4(64) | 1.25 | I | 2(32) | 0.63 | D | 4(128) | 0.75 | D | 4(16) | 1.25 | I |
| 13 | AS-35 | 8(32) | 1.00 | D | 4(32) | 0.75 | D | 16(64) | 1.25 | I | 16(64) | 1.25 | I | 4(16) | 1.25 | I |
| 14 | AS-36 | 8(64) | 1.50 | I | 4(8) | 0.38 | S | 4(64) | 0.50 | S | 16(64) | 1.25 | I | 4(16) | 1.25 | I |
| 15 | AS-37 | 4(64) | 1.25 | I | 4(16) | 0.50 | S | 4(128) | 0.75 | D | 8(16) | 0.75 | D | 4(16) | 1.25 | I |
| 16 | AS-38 | 1(8) | 0.31 | S | 4(32) | 0.75 | D | 4(64) | 0.75 | D | 8(32) | 1.00 | D | 4(64) | 1.25 | I |
| 17 | AS-39 | 1(8) | 0.31 | S | 4(64) | 1.25 | I | 4(64) | 0.75 | D | 4(64) | 1.25 | I | 4(128) | 1.25 | I |
| 18 | AS-42 | 16(64) | 1.00 | D | 8(64) | 1.25 | I | 8(128) | 0.75 | D | 8(32) | 0.75 | D | 32(256) | 2.00 | I |
| 19 | AS-47 | 16(64) | 1.00 | D | 4(64) | 1.13 | I | 8(64) | 0.50 | S | 8(64) | 1.25 | I | 16(256) | 1.50 | I |
| 20 | AS-50 | 8(64) | 1.00 | D | 8(64) | 1.50 | I | 2(32) | 0.25 | S | 2(32) | 0.63 | D | 8(128) | 1.00 | D |
| 21 | AS-51 | 4(32) | 0.50 | S | 8(32) | 1.00 | D | 2(64) | 0.63 | D | 2(4) | 0.38 | S | 32(128) | 2.50 | I |
| 22 | AS-52 | 8(64) | 0.75 | D | 8(64) | 1.25 | I | 8(64) | 0.75 | D | 4(8) | 0.63 | D | 16(128) | 1.00 | D |
| 23 | AS-54 | 8(64) | 1.00 | D | 4(16) | 0.50 | S | 8(64) | 1.00 | D | 8(4) | 0.75 | D | 8(256) | 1.50 | I |
| 24 | AZ-02 | 8(32) | 1.50 | I | 8(64) | 0.75 | D | 2(16) | 0.38 | S | 2(4) | 0.38 | S | 8(16) | 1.00 | D |
| 25 | AZ-06 | 16(16) | 1.00 | D | 8(64) | 0.50 | S | 8(32) | 0.75 | D | 4(4) | 0.38 | S | 8(256) | 1.25 | I |
| 26 | AZ-10 | 4(32) | 1.25 | I | 8(32) | 1.00 | D | 16(64) | 1.25 | I | 4(8) | 0.75 | D | 8(128) | 1.50 | I |
| 27 | AZ-25 | 8(32) | 0.75 | D | 8(64) | 1.25 | I | 4(64) | 0.63 | D | 4(16) | 0.63 | D | 4(256) | 1.13 | I |
| 28 | AZ-26 | 8(16) | 1.00 | D | 8(64) | 1.50 | I | 8(64) | 0.75 | D | 8(16) | 1.00 | D | 8(256) | 1.50 | I |
| 29 | AZ-36 | 1(32) | 1.06 | I | 8(64) | 1.50 | I | 8(64) | 0.75 | D | 8(8) | 1.00 | D | 16(128) | 2.00 | I |
| 30 | AZ-41 | 16(32) | 1.50 | I | 8(64) | 1.25 | I | 8(64) | 0.50 | S | 4(4) | 0.38 | S | 16(64) | 1.50 | I |
| 31 | AZ-42 | 16(32) | 1.50 | I | 8(64) | 1.25 | I | 8(64) | 0.50 | S | 8(8) | 0.75 | D | 8(128) | 1.25 | I |
| 32 | AZ-43 | 8(32) | 1.25 | I | 8(64) | 1.25 | I | 8(128) | 0.75 | D | 8(64) | 1.25 | I | 16(16) | 1.50 | I |
| 33 | AZ-44 | 16(32) | 1.00 | D | 8(32) | 0.75 | D | 4(64) | 0.63 | D | 8(64) | 1.25 | I | 4(16) | 1.13 | I |
| 34 | AZ-46 | 1(32) | 1.06 | I | 8(16) | 0.75 | D | 4(64) | 0.75 | D | 4(16) | 0.50 | S | 16(16) | 2.00 | I |

AS. Isolates recovered from Assuit University; AZ, Isolates recovered from Al-Azhar University; CIP, ciprofloxacin; FIC, fractional inhibitory concentration; Int. Interpretation; CIP-AMP μg/ml (MIC of ciprofloxacin-ampicillin after combination; CIP-CRO μg/ml (MIC of ciprofloxacin-ceftriaxone after combination; CIP-AK μg/ml (MIC of ciprofloxacin-amikacin after combination; CIP-DO μg/ml (MIC of ciprofloxacin-doxycycline- after combination; CIP-VC μg/ml (MIC of ciprofloxacin-vancomycin after combination; S, Synergism ≤0.5; D, Additive >0.5 ≥1; I, Indifference >1 and ≤4.0.

**Table 3. Effects of LEV combinations with different antimicrobial agents by the checkerboard method.**

| No. | Isolates code | LEV-AMP µg/ml | ΣFIC | Int | LEV-CRO µg/ml | ΣFIC | Int | LEV-AK µg/ml | ΣFIC | Int | LEV-DO µg/ml | ΣFIC | Int | LEV-VC µg/ml | ΣFIC | Int |
|---|---|---|---|---|---|---|---|---|---|---|---|---|---|---|---|---|
| 1 | AS-07 | 8(64) | 2.00 | I | 8(64) | 2.00 | I | 4(128) | 1.00 | D | 8(64) | 2.00 | I | 8(128) | 2.00 | I |
| 2 | AS-09 | 1(64) | 1.13 | I | 8(128) | 2.00 | I | 4(128) | 1.00 | D | 8(64) | 1.25 | I | 8(128) | 2.00 | I |
| 3 | AS-15 | 8(64) | 2.00 | I | 8(64 | 2.00 | I | 4(256) | 1.50 | I | 2(64) | 0.50 | S | 8(128) | 2.00 | I |
| 4 | AS-18 | 8(32) | 1.00 | D | 16(64) | 2.00 | I | 32(256) | 3.00 | I | 32(64) | 2.25 | I | 16(128) | 2.00 | I |
| 5 | AS-19 | 16(32) | 1.50 | I | 16(128) | 1.50 | I | 8(256) | 1.50 | I | 16(64) | 1.25 | I | 16(128) | 2.00 | I |
| 6 | AS-24 | 8(32) | 1.00 | D | 16(64) | 2.00 | I | 4(8) | 0.50 | S | 4(8) | 0.75 | D | 16(32) | 3.00 | I |
| 7 | AS-25 | 8(32) | 1.00 | D | 4(64) | 1.25 | I | 32(64) | 3.00 | I | 2(4) | 0.38 | S | 16(16) | 2.00 | I |
| 8 | AS-26 | 8(64) | 2.00 | I | 4(64) | 1.50 | I | 4(256) | 1.50 | I | 8(64) | 1.25 | I | 8(16) | 2.00 | I |
| 9 | AS-30 | 8(64) | 2.00 | I | 4(64) | 1.25 | I | 8(256) | 1.50 | I | 8(64) | 0.75 | D | 8(8) | 1.00 | D |
| 10 | AS-31 | 1(16) | 0.50 | S | 4(128) | 2.00 | I | 1(16) | 0.50 | S | 4(64) | 1.25 | I | 2(8) | 1.00 | D |
| 11 | AS-32 | 16(128) | 1.50 | I | 16(256) | 1.50 | I | 16(128) | 1.00 | D | 16(64) | 0.75 | D | 32(32) | 1.50 | I |
| 12 | AS-34 | 1(16) | 0.50 | S | 2(16) | 0.75 | D | 1(16) | 0.50 | S | 4(128) | 1.50 | I | 2(16) | 1.50 | I |
| 13 | AS-35 | 1(16) | 0.50 | S | 4(64) | 2.00 | I | 8(256) | 3.00 | I | 4(64) | 1.25 | I | 2(16) | 1.50 | I |
| 14 | AS-36 | 1(16) | 0.50 | S | 4(64) | 2.00 | I | 2(64) | 0.75 | D | 4(64) | 1.25 | I | 4(16) | 2.00 | I |
| 15 | AS-37 | 2(32) | 1.00 | D | 4(32) | 1.50 | I | 2(64) | 0.75 | D | 1(8) | 0.38 | S | 4(16) | 2.00 | I |
| 16 | AS-38 | 1(8) | 0.50 | S | 1(16) | 0.50 | S | 2(32) | 0.75 | D | 1(4) | 0.31 | S | 4(64) | 2.00 | I |
| 17 | AS-39 | 1(4) | 0.38 | S | 2(64) | 1.50 | I | 1(128) | 1.25 | I | 2(64) | 1.50 | I | 4(64) | 1.50 | I |
| 18 | AS-42 | 8(64) | 1.50 | I | 8(64) | 2.00 | I | 8(256) | 2.00 | I | 4(32) | 1.00 | D | 8(128) | 1.50 | I |
| 19 | AS-47 | 4(128) | 1.50 | I | 4(64) | 1.50 | I | 8(256) | 2.00 | I | 8(64) | 2.00 | I | 8(128) | 1.50 | I |
| 20 | AS-50 | 8(128) | 2.00 | I | 8(64) | 2.00 | I | 8(256) | 2.00 | I | 2(32) | 0.75 | D | 8(256) | 2.00 | I |
| 21 | AS-51 | 8(128) | 2.00 | I | 8(64) | 2.00 | I | 4(128) | 1.50 | I | 1(4) | 0.38 | S | 32(256) | 5.00 | A |
| 22 | AS-52 | 16(128) | 2.00 | I | 8(64) | 1.50 | I | 8(128) | 1.50 | I | 2(8) | 0.63 | D | 16(256) | 2.00 | I |
| 23 | AS-54 | 8(128) | 2.00 | I | 4(64) | 1.50 | I | 8(128) | 2.00 | I | 2(4) | 0.50 | S | 8(256) | 2.00 | I |
| 24 | AZ-02 | 4(32) | 1.50 | I | 8(128) | 1.50 | I | 2(16) | 0.50 | S | 2(2) | 0.38 | S | 4(16) | 1.00 | D |
| 25 | AZ-06 | 4(16) | 1.00 | D | 8(128) | 1.50 | I | 2(16) | 0.50 | S | 2(8) | 0.75 | D | 8(256) | 2.00 | I |
| 26 | AZ-10 | 8(32) | 2.00 | I | 8(64) | 2.00 | I | 16(256) | 3.00 | I | 4(8) | 1.00 | D | 8(128) | 2.00 | I |
| 27 | AZ-25 | 1(8) | 0.38 | S | 8(128) | 4.00 | I | 4(128) | 2.00 | I | 4(16) | 1.50 | I | 4(256) | 2.00 | I |
| 28 | AZ-26 | 8(16) | 1.50 | I | 8(128) | 3.00 | I | 8(256) | 2.00 | I | 8(16) | 1.50 | I | 8(256) | 2.00 | I |
| 29 | AZ-36 | 16(32) | 3.00 | I | 8(64) | 2.00 | I | 8(256) | 2.00 | I | 4(8) | 1.00 | D | 16(128) | 3.00 | I |
| 30 | AZ-41 | 1(8) | 0.50 | S | 1(16) | 0.50 | S | 2(128) | 1.00 | D | 1(4) | 0.50 | S | 4(64) | 2.00 | I |
| 31 | AZ-42 | 8(32) | 2.00 | I | 8(64) | 2.00 | I | 8(256) | 2.00 | I | 2(4) | 0.50 | S | 8(128) | 2.00 | I |
| 32 | AZ-43 | 1(8) | 0.50 | S | 1(16) | 0.50 | S | 8(256) | 3.00 | I | 8(64) | 3.00 | I | 2(16) | 1.50 | I |
| 33 | AZ-44 | 2(32) | 1.00 | D | 4(64) | 2.00 | I | 4(128) | 2.00 | I | 8(64) | 3.00 | I | 2(8) | 1.00 | D |
| 34 | AZ-46 | 16(32) | 2.00 | I | 8(32) | 1.00 | D | 8(128) | 1.50 | I | 4(32) | 0.75 | D | 16(16) | 2.00 | I |

AS. Isolates recovered from Assuit University; AZ, Isolates recovered from Al-Azhar University; LEV, levofloxacin; FIC, fractional inhibitory concentration; Int. Interpretation; LEV-AMP µg/ml (MIC of levofloxacin—ampicillin after combination); LEV-CRO µg/ml (MIC of levofloxacin -ceftriaxone after combination); LEV-AK µg/ml (MIC of levofloxacin—amikacin after combination; LEV-DO µg/ml (MIC of levofloxacin—doxycycline after combination); LEV-VC µg/ml (MIC of levofloxacin—vancomycin after combination); S, Synergism ≤0.5; D, Additive >0.5 ≥1; I, Indifference >1 and ≤4.0; A, Antagonism >4.

susceptibility patterns of 100 *A. baumannii* isolates were determined revealing a high resistance to most of the tested antimicrobial agents (76–99%). On the other hand, colistin was the most effective anti-microbial agent against *A. baumannii* 95% followed by doxycycline 43%. Our study agrees with the previous study which recorded that, *A. baumannii* pathogen as an opportunistic pathogen often susceptible to colistin and having a low susceptibility to other antimicrobial agents causing radical morbidity and mortality [35]. The non-rational use of

**Table 4. Summary of antibiotic combinations with FQs by checkerboard method on antibiotic resistant isolates.**

| Antibiotics combination | MICs range of FQs after addition other antibiotics | ΣFIC Range | Activity | | | |
|---|---|---|---|---|---|---|
| | | | Synergy No. (%) | Additive No. (%) | Indifference No. (%) | Antagonist No. (%) |
| CIP-AMP | 1–32 | 0.31–1.5 | 8(23.53) | 14(41.18) | 12(35.29) | 0 |
| LEV-AMP | 1–16 | 0.38–3 | 9(26.47) | 6(17.65) | 19(55.88) | 0 |
| CIP-CRO | 4–16 | 0.38–2 | 6(17.65) | 11(32.35) | 17(50) | 0 |
| LEV-CRO | 1–16 | 0.5–4 | 3(8.28) | 2(5.88) | 29(85.29) | 0 |
| CIP-AK | 2–32 | 0.25–2.5 | 11(32.35) | 18(52.94) | 5(14.71) | 0 |
| LEV-AK | 1–32 | 0.5–3 | 5(14.71) | 7(20.59) | 22(64.71) | 0 |
| CIP-DO | 2–32 | 0.38–2.25 | 6(17.65) | 20(58.82) | 8(23.53) | 0 |
| LEV-DO | 1–32 | 0.31–3 | 9(26.47) | 10(29.41) | 15(44.12) | 0 |
| CIP-VC | 2–32 | 0.75–3 | 0 | (17.65)6 | (82.35)28 | 0 |
| LEV-VC | 2–32 | 1–5 | 0 | 29(85.29) | 4(11.76) | 1(2.94) |

FQs: fluoroquinolones; MICs: minimum inhibitory concentrations; FIC, fractional inhibitory concentration; CIP: ciprofloxacin, LEV: levofloxacin; AMP: ampicillin; CRO: ceftriaxone; AK: Amikacin; DO: doxycycline; VC: vancomycin.

antimicrobial agents may be considered as the main cause for resistance development of this harmful pathogen.

The XDR and PDR isolates of *A. baumannii* are a leading cause of hospital-acquired infections [39]. PDR and XDR are increasingly being reported worldwide in clinical [40] or environmental isolates [41]. In the present study, 2, 68 and 30% of the recovered isolates exhibited PDR, XDR and MDR phenotypes, respectively. Phenotypic analysis of their susceptibility to the 19 antimicrobial agents showed the diversity of the resistance of these isolates. They were divided into 12 major profiles according to the number of antimicrobial agents to which they were resistant. It was previously reported that, the high prevalence of MDR, XDR and PDR *A. baumannii* was due to the misuse of the antimicrobial agents [42, 43], or due to the differences in rates of infections by the respective pathogen (which could be an indications of how strictly the hygiene protocols and good manufacturing practice are applied in different hospitals) [44], in addition to the plasticity and endless capacity of *A. baumannii* genomes [45].

The use of ERIC-PCR as a genotyping technique to inspect the epidemics of hospital-acquired infection depends on its ability to epidemiologically relate the collected isolates during a nosocomial outbreak and investigate if the involved isolates are genetically related or originated from different strains [46, 47]. In our study, the phylogenetic dendrogram of ERIC-PCR showed that the isolates can be divided into three major clusters. The respective diversity of the collected isolates into three major clusters may be due to the multiple contamination sources of *A. baumannii*. This result could be an indication of clonal expansion and microbial colonization from different sources as previously reported [48]. Therefore, the obtained results emphasized the urgent need for a new scenario of drug combinations, new therapeutics options, as well as accelerating the development of the new infection control strategies to combat MDR resistance particularly of clinically relevant pathogens such as *A. baumannii*.

Little information is available on successful treatment regimens of *A. baumannii* by FQs combinations with antimicrobial or non-antimicrobial agents. Few attempts have been made so far to control the resistance of this strain [10, 49]. Our results revealed that both doxycycline and colistin still retain good activities against *A. baumannii*. However, both antibiotics have not been recommended in the treatment of severe infections caused by *A. baumannii*, particularly in the intensive care unit and cardiac care unit [50]. Several studies have been conducted

**Table 5. Effects of non-antibiotics on the MIC of CIP.**

| No. | Isolates code | MICs (µg/ml) CIP alone | CPZ 200µg/ml | | PR 0.5 mg /ml | | PR 1mg /ml | | DIC 4mg/ ml | |
|---|---|---|---|---|---|---|---|---|---|---|
| | | | MICs (µg/ml) CIP +CPZ | MDF | MICs (µg/ml) CIP +PR | MDF | MICs (µg/ml) CIP +PR | MDF | MICs (µg/ml) CIP + DIC | MDF |
| 1 | AS-07 | 16 | 8 | 2 | 0.5 | 32 | 0.03 | 533.3 | 0.25 | 64 |
| 2 | AS-09 | 32 | 2 | 16 | 0.5 | 64 | 0.03 | 1066.7 | 0.03 | 1066.7 |
| 3 | AS-15 | 16 | 8 | 2 | 4 | 4 | 0.06 | 266.7 | 0.125 | 128 |
| 4 | AS-18 | 32 | 8 | 4 | 4 | 8 | 0.25 | 128 | 0.125 | 256 |
| 5 | AS-19 | 32 | 8 | 4 | 4 | 8 | 0.25 | 128 | 0.125 | 256 |
| 6 | AS-24 | 32 | 8 | 4 | 0.5 | 64 | 0.03 | 1066.7 | 0.125 | 256 |
| 7 | AS-25 | 16 | 2 | 8 | 1 | 16 | 0.06 | 266.7 | 0.03 | 533.3 |
| 8 | AS-26 | 32 | 2 | 16 | 1 | 32 | 0.06 | 533.3 | 0.125 | 256 |
| 9 | AS-30 | 32 | 2 | 16 | 1 | 32 | 0.06 | 533.3 | 0.06 | 533.3 |
| 10 | AS-31 | 8 | 4 | 2 | 0.125 | 64 | 0.03 | 266.7 | 0.03 | 266.7 |
| 11 | AS-32 | 64 | 16 | 4 | 8 | 8 | 0.25 | 256 | 0.125 | 512 |
| 12 | AS-34 | 16 | 8 | 2 | 2 | 8 | 0.03 | 533.3 | 0.03 | 533.3 |
| 13 | AS-35 | 16 | 8 | 2 | 8 | 2 | 0.06 | 266.7 | 0.06 | 266.7 |
| 14 | AS-36 | 16 | 8 | 2 | 8 | 2 | 0.06 | 266.7 | 0.03 | 533.3 |
| 15 | AS-37 | 16 | 8 | 2 | 8 | 2 | 0.06 | 266.7 | 0.06 | 266.7 |
| 16 | AS-38 | 16 | 8 | 2 | 0.5 | 32 | 0.03 | 533.3 | 0.06 | 266.7 |
| 17 | AS-39 | 16 | 8 | 2 | 2 | 8 | 0.06 | 266.7 | 0.06 | 266.7 |
| 18 | AS-42 | 32 | 4 | 8 | 1 | 32 | 0.25 | 128 | 0.06 | 533.3 |
| 19 | AS-47 | 32 | 0.5 | 64 | 0.5 | 64 | 0.03 | 1066.7 | 0.125 | 256 |
| 20 | AS-50 | 16 | 8 | 2 | 8 | 2 | 0.06 | 266.7 | 0.03 | 533.33 |
| 21 | AS-51 | 16 | 1 | 16 | 1 | 16 | 0.25 | 64 | 0.125 | 128 |
| 22 | AS-52 | 32 | 1 | 32 | 1 | 32 | 0.25 | 128 | 0.03 | 1066.7 |
| 23 | AS-54 | 16 | 1 | 16 | 0.5 | 32 | 0.03 | 533.3 | 0.125 | 128 |
| 24 | AZ-02 | 16 | 1 | 16 | 0.5 | 32 | 0.03 | 533.3 | 0.125 | 128 |
| 25 | AZ-06 | 32 | 0.5 | 64 | 0.5 | 64 | 0.03 | 1066.7 | 0.125 | 256 |
| 26 | AZ-10 | 16 | 0.5 | 32 | 0.5 | 32 | 0.03 | 533.3 | 0.06 | 266.67 |
| 27 | AZ-25 | 32 | 0.125 | 256 | 4 | 8 | 0.06 | 533.3 | 0.125 | 256 |
| 28 | AZ-26 | 16 | 0.125 | 128 | 4 | 4 | 0.03 | 533.3 | 0.03 | 533.3 |
| 29 | AZ-36 | 16 | 0.125 | 128 | 0.125 | 128 | 0.06 | 266.7 | 0.03 | 533.3 |
| 30 | AZ-41 | 32 | 0.125 | 256 | 4 | 8 | 0.125 | 256 | 0.125 | 256 |
| 31 | AZ-42 | 32 | 0.125 | 256 | 4 | 8 | 0.125 | 256 | 0.125 | 256 |
| 32 | AZ-43 | 32 | 0.125 | 256 | 4 | 8 | 0.03 | 1066.7 | 0.125 | 256 |
| 33 | AZ-44 | 32 | 0.125 | 256 | 2 | 16 | 0.03 | 1066.7 | 0.125 | 256 |
| 34 | AZ-46 | 16 | 0.125 | 128 | 4 | 4 | 0.03 | 533.3 | 0.03 | 533.3 |

AS. Isolates recovered from Assuit University; AZ, Isolates recovered from Al-Azhar University; MICs, minimum inhibitory concentrations; MDF, MIC decrease factor; CIP, ciprofloxacin; CPZ, chlorpromazine; PR. Propranolol; DIC, diclofenac.

to evaluate various antibiotic combinations to be used as options for the treatment of MDR *A. baumannii* clinical isolates. However, the results obtained were controversial and were attributed to many other factors [51–53]. In this study, a new scenario has been attempted for finding possible therapeutic options to control infections caused by XDR- and PDR- *A. baumannii* clinical isolates. This was carried out through an evaluation of the use of CIP or LEV, the two most widely used antimicrobial agents for the treatment of *A. baumannii*-associated infections, in combination with either antibiotics or non- antibiotics including, PR, DIC, and CPZ.

**Table 6. Effects of non -antibiotic combinations on the MIC of LEV.**

| No. | Isolates code | MICs (µg/ml) LEV | CPZ 200µg/ml | | PR 0.5 mg /ml | | PR 1mg /ml | | DIC 4mg/ ml | |
|---|---|---|---|---|---|---|---|---|---|---|
| | | | MICs (µg/ml) LEV +CPZ | MDF | MICs (µg/ml) LEV +PR | MDF | MICs (µg/ml) LEV +PR | MDF | MICs (µg/ml) LEV +DIC | MDF |
| 1 | AS-07 | 8 | 2 | 4 | 0.5 | 16 | 0.03 | 266.7 | 0.125 | 64 |
| 2 | AS-09 | 8 | 0.5 | 16 | 0.5 | 16 | 0.03 | 266.7 | 0.03 | 266.7 |
| 3 | AS-15 | 8 | 0.25 | 32 | 4 | 2 | 0.06 | 133.3 | 0.06 | 133.3 |
| 4 | AS-18 | 16 | 0.5 | 32 | 4 | 4 | 0.125 | 128 | 0.06 | 266.7 |
| 5 | AS-19 | 16 | 0.5 | 32 | 4 | 4 | 0.125 | 128 | 0.06 | 266.7 |
| 6 | AS-24 | 16 | 0.5 | 32 | 0.5 | 32 | 0.03 | 533.3 | 0.03 | 533.3 |
| 7 | AS-25 | 16 | 0.06 | 266.7 | 1 | 16 | 0.06 | 266.7 | 0.03 | 533.3 |
| 8 | AS-26 | 8 | 0.5 | 16 | 1 | 8 | 0.06 | 133.3 | 0.25 | 32 |
| 9 | AS-30 | 16 | 0.25 | 64 | 1 | 16 | 0.06 | 266.7 | 0.06 | 266.7 |
| 10 | AS-31 | 4 | 0.125 | 32 | 0.125 | 32 | 0.03 | 133.3 | 0.03 | 133.3 |
| 11 | AS-32 | 32 | 8 | 4 | 4 | 8 | 0.125 | 256 | 0.125 | 256 |
| 12 | AS-34 | 4 | 0.5 | 8 | 2 | 2 | 0.03 | 133.3 | 0.03 | 133.3 |
| 13 | AS-35 | 4 | 0.5 | 8 | 2 | 2 | 0.06 | 66.7 | 0.06 | 66.7 |
| 14 | AS-36 | 4 | 1 | 4 | 2 | 2 | 0.06 | 66.7 | 0.03 | 133.3 |
| 15 | AS-37 | 4 | 1 | 4 | 1 | 4 | 0.06 | 66.7 | 0.06 | 66.7 |
| 16 | AS-38 | 4 | 1 | 4 | 0.5 | 8 | 0.03 | 133.3 | 0.06 | 66.7 |
| 17 | AS-39 | 4 | 1 | 4 | 2 | 2 | 0.06 | 66.7 | 0.06 | 66.7 |
| 18 | AS-42 | 8 | 2 | 4 | 1 | 8 | 0.03 | 266.7 | 0.06 | 133.3 |
| 19 | AS-47 | 8 | 0.5 | 16 | 0.5 | 16 | 0.03 | 266.7 | 0.125 | 64 |
| 20 | AS-50 | 8 | 8 | 1 | 1 | 8 | 0.06 | 133.3 | 0.03 | 266.7 |
| 21 | AS-51 | 8 | 0.25 | 32 | 1 | 8 | 0.125 | 64 | 0.125 | 64 |
| 22 | AS-52 | 16 | 0.25 | 64 | 4 | 4 | 0.125 | 128 | 0.03 | 533.3 |
| 23 | AS-54 | 8 | 0.25 | 32 | 0.5 | 16 | 0.03 | 266.7 | 0.06 | 133.3 |
| 24 | AZ-02 | 8 | 0.25 | 32 | 0.5 | 16 | 0.03 | 266.7 | 0.03 | 266.7 |
| 25 | AZ-06 | 8 | 0.125 | 64 | 0.5 | 16 | 0.03 | 266.7 | 0.06 | 133.3 |
| 26 | AZ-10 | 8 | 0.25 | 32 | 0.5 | 16 | 0.03 | 266.7 | 0.06 | 133.3 |
| 27 | AZ-25 | 4 | 0.125 | 32 | 1 | 4 | 0.06 | 66.7 | 0.06 | 66.7 |
| 28 | AZ-26 | 8 | 0.125 | 64 | 2 | 4 | 0.03 | 266.7 | 0.03 | 266.7 |
| 29 | AZ-36 | 8 | 0.125 | 64 | 0.06 | 133.3 | 0.03 | 266.7 | 0.03 | 266.7 |
| 30 | AZ-41 | 4 | 0.125 | 32 | 0.03 | 133.3 | 0.03 | 133.3 | 0.125 | 32 |
| 31 | AZ-42 | 8 | 0.125 | 64 | 2 | 4 | 0.125 | 64 | 0.06 | 133.3 |
| 32 | AZ-43 | 4 | 0.125 | 32 | 0.5 | 8 | 0.03 | 133.3 | 0.125 | 32 |
| 33 | AZ-44 | 4 | 0.125 | 32 | 0.5 | 8 | 0.03 | 133.3 | 0.06 | 66.7 |
| 34 | AZ-46 | 16 | 0.125 | 128 | 1 | 16 | 0.03 | 533.3 | 0.03 | 533.3 |

AS. Isolates recovered from Assuit University; AZ, Isolates recovered from Al-Azhar University; MICs, minimum inhibitory concentrations; MDF, MIC decrease factor; LEV, levofloxacin; CPZ, chlorpromazine; PR. Propranolol; DIC, diclofenac.

In our study, 34 *A. baumannii* clinical isolates were selected based on their different antibiotic resistance profiles as well as results of ERIC-PCR that revealed that these isolates were not clonal. Drug combinations of either CIP or LEV with ampicillin, ceftriaxone, amikacin, doxycycline or vancomycin tested on the 34 isolates (including 2 PDR and 32 XDR) showed synergism in 23.53, 17.65, 32.35, 17.65 and 26.47, 8.28, 14.71, 26.47%, respectively. Several previous in vitro studies reported synergic effects between diverse drugs for the treatment of *A. baumannii* [51–53]. The results of the previous studies of drug combinations were classified as

**Table 7. Summary of CIP and LEV combinations with non-antibiotics.**

| Non-antibiotic | FQs(MIC before, µg/ml) | | No. (% Susceptible) by | |
|---|---|---|---|---|
| | CIP (8–64) | LEV (4–32) | CIP | LEV |
| MIC range (µg/ml) after CPZ 200 µg/ml | 0.125–16 | 0.06–8 | 15(44.12) | 32(94.12) |
| MIC range (µg /ml) after PR 0.5 mg/ml | 0.125–8 | 0.03–4 | 17(50) | 29(85.29) |
| MIC range (µg/ml) after PR 1 mg/ml | 0.03–0.25 | 0.03–0.125 | 34(100) | 34(100) |
| MIC range (µg/ml) after sodium dilofenac 4 mg/ml | 0.03–0.125 | 0.03–0.25 | 34(100) | 34(100) |

FQs: fluoroquinolones, MIC: minimum inhibitory concentration, CIP: ciprofloxacin, LEV: levofloxacin, CPZ: Chlorpromazine.

synergistic, additive, indifferent, or even antagonistic. FQs are major antimicrobial agents associated with induction of resistance, but when used in combination, may prevent resistance [54, 55]. Our study revealed that, AMP reduced MICs of CIP (8–64 µg/ml) and LEV (4–32 µg/ml) to 1–32 µg/ml and 1–16 µg/ml, enhancing their efficacy by 23.53% and 26.47%, respectively. However, these results do not agree with a previous study that was conducted in north Egypt [9]. This contradiction could be due to intrinsic resistance, genetic differentiation of the isolates as well as the geographical factor. CRO also reduced MICs of CIP and LEV and showed a synergistic effect by 17.65 and 8.28%, respectively. These results agree with a previous study that reported successful combinations of CIP with cephalosporins against *Pseudomonas* spp. [56]. In the present study, successful combination of AMK with CIP and LEV were obtained, which reduced MICs to 2–32 and 1–32 µg/ml with synergies of 32.35 and 14.71%, respectively. Our results are in accordance with a previous study conducted in Upper Egypt, which reported a successful CIP-AMK combination against Gram-negative pathogens [10]. It was previously stated that, amikacin combination with FQs is used to expand the antimicrobial spectrum, reduce toxicity, and prevent or diminish the emergence of resistant mutants of FQs [57].

Different mechanisms in *A. baumannii* isolates were reported to be responsible to resistance to beta-lactams and other important classes of antibiotics, leading to the emergence of PDR *A. baumannii* causing nosocomial infections [58]. Doxycycline is a bacteriostatic antibiotic and showed high activity against *A. baumannii* either in monotherapy or combinations [11, 59]. The DO-FQs combination reduced the MICs of CIP and LEV from 8–64 and 4–32 µg/ml to 2–32 and 1–32 µg/ml, respectively. Our results agreed with previous studies which showed the ability of doxycycline to potentiate the efficacy of other antimicrobial agents when used in combination with FQs [60, 61]. Although vancomycin acts by hindering cell wall synthesis in the Gram-positive pathogens, it showed a synergistic effect when used in combination with colistin against *A. baumannii* [11]. Moreover, the membrane-permeabilizing properties of colistin was reported to enhance the activity of vancomycin against *A. baumannii* [11]. In this study, in vitro combinations of vancomycin with two members of FQs, the CIP and LEV did not show any synergistic effect against PDR *A. baumannii* clinical isolates.

Several successful combinations of FQs with natural products against MDR *A. baumannii* were previously reported [62, 63]. However, few studies were conducted on the use of some non-selective beta-blockers such as PR in combination with antimicrobial agents [7] or carvedilol alone [14]. In the present study, combination of CIP or LEV with PR successfully overcame bacterial resistance of XDR and PDR *A. baumannii* clinical isolates. CIP or LEV combination with PR at concentration 0.5 mg/ml significantly reduced the MICs of CIP and LEV and at 1 mg/ml completely inhibited the resistance of *A. buamannii*. The effect of PR as an efflux pump inhibitor [64], or as an antibacterial [65] has been investigated. Different chemotherapeutics are nowadays known to inhibit or diminish the microbial resistance of XDR

and PDR of different microorganisms. Various drugs have got consideration recently in different research fields, including cancer therapy as a transporter and drug delivery as PR [13, 66], calcium channel blockers [67] natural products such as reserpine [68]. In this study, a combination of CIP or LEV with DIC at 4 mg/ml restored the susceptibility of tested XDR, and PDR *A. baumannii* to FQs. Our results fully agreed with Dutta et al which showed successful synergism of DIC with antimycobacterial drugs [69]. Furthermore, antipyretics and NSAIDs are commonly co-administered with antimicrobial therapy and primarily act by inhibiting prostaglandin synthesis. Although their exact function is uncertain, several suggestions have been explained [70, 71]. They include changing the surface hydrophobicity of microbes [18], altering the expression of virulence factor, influencing biofilm production [19], affecting motility and metabolism, inhibiting quorum sensing among microbes [72], interacting with the transport and release of antibiotics by polymorphonuclear leukocytes (PMNL) [20], and modifying the susceptibility of microbes to antimicrobial therapy [15–17]. The effect of NSAIDs on the antibiotic susceptibility of pathogens was investigated which mostly resulted from a change in direct antimicrobial penetration through cell membranes of bacteria or from an increase or decrease in efflux through the membranes [15, 16, 73]. By understanding these mechanisms, these synergistic effects can be exploited in the treatment of infectious diseases and potential compromising effects on antimicrobial efficacy can be avoided [15]. However, decreased susceptibility can also result from induced β-lactamase activity [74]. DIC has analgesic, antipyretic, as well as anti-inflammatory characters. This non-steroidal anti-inflammatory drug has demonstrated strong antimicrobial property when tested against a large number of bacteria and has bactericidal activity in nature due to inhibition of DNA synthesis [5, 75].

Antipsychotics of different groups such as phenothiazines (CPZ hydrochloride) have significant antibacterial activity [6, 76] and act as efflux pump inhibitors [77]. In this study, CPZ combinations with CIP or LEV have markedly reduced the MIC of resistance isolates. Our result agreed with studies previously carried out against *Mycobacterium spp* [78, 79]. It has been found that CPZ had a significant bactericidal effect [6, 75] in addition to the efflux pump inhibitor action when tested for their effect on antibiotic resistance [79, 80].

## Conclusion

Several combination regimens have been successfully evaluated in vitro for combating antimicrobial resistance of PDR and XDR isolates. In this research, high prevalence of PDR and XDR *A. baumannii* isolates associated with nosocomial infections in Upper Egypt was observed and therefore, exerted a negative impact on patient health and disease prognosis. Propranolol, chlorpromazine and diclofenac restore susceptibility of some selected XDR *A. baumannii* to CIP and LEV. More pharmacokinetic/pharmacodynamics studies are needed to guide the use of these combinations against these life-threatening pathogens.

## Supporting information

**S1 Fig.**
(PDF)

## Acknowledgments

The authors are grateful to the Medical Research Center, Faculty of Medicine Assiut University, and Al Azhar University hospitals for providing *A. buamannii* isolates. More grateful is extended to Prof. Enas A Deaf, Prof. Nahla M Elsherbiny, and Assist. Prof. Mohamed A. EL-Mokhtar; Department of microbiology and immunology, Faculty of Medicine, Assiut

University, and Dr. Mohamed A. Abdel-Lateef; Department of Pharmaceutical Analytical Chemistry, Faculty of Pharmacy, Al-Azhar University, Assiut branch for their helping during the study.

## Author Contributions

**Conceptualization:** Mostafa A. Mohammed, Mohammed T. Ahmed, Bahaa E. Anwer, Khaled M. Aboshanab, Mohammad M. Aboulwafa.

**Data curation:** Mostafa A. Mohammed, Bahaa E. Anwer.

**Formal analysis:** Mohammed T. Ahmed, Khaled M. Aboshanab, Mohammad M. Aboulwafa.

**Investigation:** Mostafa A. Mohammed, Mohammad M. Aboulwafa.

**Methodology:** Mostafa A. Mohammed, Mohammed T. Ahmed, Bahaa E. Anwer, Khaled M. Aboshanab.

**Supervision:** Khaled M. Aboshanab, Mohammad M. Aboulwafa.

**Writing – original draft:** Mostafa A. Mohammed, Bahaa E. Anwer.

**Writing – review & editing:** Mohammed T. Ahmed, Khaled M. Aboshanab.

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
