## [Decision Letter · Decision Letter 0]

2 Jun 2020

PONE-D-20-04012

Propranolol and chlorpromazine restore susceptibility of extensively drug-resistant (XDR)-Acinetobacter baumannii to fluoroquinolones

PLOS ONE

Dear Dr. Aboshanab,

Thank you for submitting your manuscript to PLOS ONE. After careful consideration, we feel that it has merit but does not fully meet PLOS ONE’s publication criteria as it currently stands. Therefore, we invite you to submit a revised version of the manuscript that addresses the points raised during the review process. Please focus on the suggestions provided by reviewer 1.

We look forward to receiving your revised manuscript.

Kind regards,

Monica Cartelle Gestal, PhD

Academic Editor

PLOS ONE

Journal Requirements:

Additional Editor Comments (if provided):

Dear Aboshanab,

Thanks for considering publishing with Plos One. Although very interesting, we request major revisions prior to further evaluation of the manuscript. Please address the comments and suggestions provided by the reviewers and specially those suggested by Reviewer 1.

Reviewers' comments:

Reviewer's Responses to Questions

**Comments to the Author**

1. Is the manuscript technically sound, and do the data support the conclusions?

Reviewer #1: Partly

Reviewer #2: Yes

2. Has the statistical analysis been performed appropriately and rigorously? 

Reviewer #1: No

Reviewer #2: N/A

3. Have the authors made all data underlying the findings in their manuscript fully available?

Reviewer #1: Yes

Reviewer #2: No

4. Is the manuscript presented in an intelligible fashion and written in standard English?

Reviewer #1: No

Reviewer #2: No

5. Review Comments to the Author

Reviewer #1: MANUSCRIPT PONE-D-20-04012

Manuscript PONE-D-20-04012 describes the effect of different drug combinations with ciprofloxacin and levofloxacin on the susceptibility of PDR and XDR A. baumannii to the fluoroquinolones CIP and LEV. Propranolol and diclofenac sodium particularly obliterated the resistance of the PDR and XDR strains. No doubt, any study reporting alternative therapies in the treatment of infections caused by A. baumannii, a species of bacteria notorious for causing difficult to treat infections in immune-compromised patients is important judging from the current global scourge antimicrobial resistance. Unfortunately, not minding the importance of the area of focus of this manuscript, it is difficult for me to recommend this paper for publication in PLOS ONE in its present form.

1. The authors did not make judicious use of existing literature to provide adequate background for their study and relate the findings to previous studies in this area.

2. They also did not follow the PLOS ONE guidelines for authors in the preparation of their manuscript. For example, PLOS ONE do not use structured abstract as they did in this manuscript. Similarly, the figure titles were not inserted in the manuscript as spelt out in the authors guidelines

3. The manuscript was not line-numbered, making reference to specific sections difficult in the course of revising the manuscript.

4. The writing style and language use need considerable improvement as the present presentation is unclear making it difficult to follow the thoughts and ideas that the authors are trying to present in some parts of the manuscript. There are important word omissions, use of the noun resistant in place of the adjectival form of the word, sentences that do not read well etc.

5. Authors should please define all abbreviations at first use

6. Most important, I am concerned about the number and narrow geographical origin of the isolates of A. baumannii (34) included in the evaluation of drug combination part of the experiment. In my own opinion, this number is not enough to make the broad conclusion reached by the authors in this study about the effects of those combined drug in ameliorating the resistance of the isolates to antimicrobial drugs.

Specific comments for the authors

Abstract

Methods section

1. Method section not detailed enough leaving out important details about the drug combination trial which is supposed to be the main focus of the manuscript. This critical omission made some of the results presented in the results section of the structured abstract orphans and unlinked to the methods section.

Results section

1. This section contained information that could have been included in the methods section of the abstract.

2. What informed the choice of the 34 isolates included in the drug combination trial? It will be important to know how the authors selected the 34 isolates, especially the 32 XDR isolates from among the 68 isolates that were XDR since the authors found out that the isolates are not clonal through ERIC-PCR. Why did they not include any of the MDR isolates in this test? Including as many isolates as possible will certainly have strengthened and validated the findings of this study (see comment no.6 above).

3. ........including 2 PDR and 32 XDR showed synergism in 23.5%, 17.65%............ : I could not understand what the authors are trying to pass across to readers here. Do they mean that the drug combination showed synergism in 23.5% etc. of the isolates? Same goes for the combination with CPZ and propranolol.

4. Does the concentration range after propranolol (0.5-1mg/ml) refer to both CPZ and propranolol?

5. The authors concluded that their results will greatly help physicians for (sic) the proper choice of antibiotic combinations for combating A. baumannii but some of the drugs combined such as CPZ and Propranolol are not antibiotics.

Keywords

1. extensively drug resistant should be qualified

Introduction

1. The information presented in this section to provide a background upon which this study is built is scanty. Authors should please add more recent published information to this section to provide a more robust background for their study and to set the study within the context of previous studies in the area.

2. Line 3-4: A. baumannii is not a hospital acquired infection but a bacterium that cause HAI

Methods

Bacterial isolates

1. What did the authors do with the incubated plates? This question becomes important in the light of the next section where they reported carrying out phenotypic and genotypic tests to identify the colonies. How many colonies did the authors subject to this identification step? From which of the three different media used did they selected the colonies? How did they prevent the selection of clonal replicates from the three agar media?

2. Can the authors supply the name of the positive control used in addition to the ATCC accession number?

Antimicrobial susceptibility testing

1. The authors mentioned colistin among the antibiotics used for this test by disc diffusion. This method is not recommended for testing of colistin susceptibility because of the large molecular weight of colistin which hinders its diffusion

2. It should not be MIC of XDR and PDR isolates against CIP and LEV, rather it is the MIC of CIP and LEV against the bacterial isolates.

3. The authors should supply the range of concentrations used for MIC determination

4. The authors mentioned vancomycin among the antibiotics used for MIC determination. Vancomycin is not recommended for use against gram negative bacteria.

Molecular typing of recovered isolates

1. In the abstract section, the authors mentioned that all isolates were not clonal while in this section, it appears that only 34 isolates were subjected to ERIC-PCR. The authors need to clarify this to let us know if they subjected all or 34 isolates to ERIC-PCR

Evaluation of drug combinations

1. The authors should please provide a brief summary of how the evaluation of drug combinations was carried out. For example how were the drugs combined? Was the combination carried out in-vitro or in-vivo? These are critical details needed to interpret the results

2. CPZ was conspicuously missing in the list of drugs evaluated in this section. Interestingly, the main conclusion of the manuscript centered on CPZ and propranolol .

Results

Specimen collection and identification of recovered A. baumannii isolates

1. The authors mentioned isolating 623 non-lactose fermenters and 977 lactose fermenters of which a total of 151 isolates were phenotypically identified Acinetobacter isolates (sic). It is important to clarify if the 151 isolates identified as Acinetobacter were a subset of the non-lactose fermenters or a subset of the lactose fermenters?

Antimicrobial susceptibility testing

1. The level of resistance to doxycycline was reported as 47% in the abstract but was reported as 57% here. Authors should please clarify which is the correct figure.

2. I am confused that the authors are referring to multiple contamination sources of A. baumannii in Upper Egypt. I am not sure of how this relate to the isolation of the bacteria of the present study from clinical specimen.

MIC

1. In the methods section, the authors mentioned that they determined the MIC using 7 antimicrobial drugs namely CIP, LEV, ampicillin, ceftriaxone, amikacin, doxycycline, and vancomycin. Curiously in the results section, results for only 2 (CIP and LEV) were given. The authors should please supply results for the remaining 5 antibiotics used in MIC determination.

Genotyping

1. Again, it becomes very important for the authors to clarify how many isolates were genotyped in the methods section. The authors made references to “obtained isolates” and “% of total isolates”. To my own understanding 100 isolates were obtained in the study, but the information here showed only 34 isolates were genotyped which does not represent all the obtained isolates or the total isolates as alluded to by the authors in this section.

Evaluation of drug combinations

1. Two antimicrobial profiles selected including...... Selected for what?

2. Table 1- I think before we can conclude on whether there is synergy or antagonism and the extent of the synergy as reported in this table, there is need for baseline references (i.e. the baseline MIC for CIP and LEV before addition of the combination antibiotics) for each isolate. That I did not see in the table. The authors should kindly correct me if I am wrong.

3. The authors should provide legends to define all abbreviations in the tables and figures.

4. Table 2- Please refer to comment no2 above. Table 2 is more like what is expected. One wonders why Table 1 is completely different from Table 2 even though they are used to report similar results.

5. The results presented by the authors in Table 2 and the manuscript showed that propranolol and diclofenac sodium completely reduced the MIC of CIP and LEV below the susceptibility cut-off value. One is therefore curious as to the reason(s) why the authors decided to lay emphasis on CPZ in the title without any mention of diclofenac sodium.

6. Labels on the axes of some of the figures are not expressing any meaning. Both axes of figure 5 carry exactly the same label.

Discussion

1. Again, the discussion of results is not clear and the authors did not use existing literature to place their result in proper perspective.

2. If the authors are reporting a total of 100 A. baumannii clinical isolates and 61 were from RTI, UTI and blood, where did the remaining 39 isolates came from?

References

1. The guidelines for authors as regards reference listing was not followed by the authors

2. The doi provided for reference no 18 is not correct.

Reviewer #2: 1. The manuscript has several typographical errors and English should be reviewed.

a) Page 4, line 2: the coma should be after "progression" not before.

b) Page 4, line 16: delete the point after "Catheters".

c) Page 4, line 24: missing space between "hours" and the reference "[10]".

d) Page 6, line 14: missing space between "guidelines" and the reference "[15]".

e) Page 7, line 19: delete the point before "(figure 1)".

f) Page 10, line 3: delete the semicolon after "CPZ".

g) Page 10, line 3: the second "CIP" should be changed by "LEV".

h) Page 10, line 4: omit semicolon after "44.12%".

i) Page 11, line 11: delete double space in "and mortality".

j) Page 11, line 12: delete double space in "due to misuse".

k) Page 11, line 19: misspelling of "originating" it should be "originated"

l) Page 12, line 1: include a reference to figure 5 when the three major clusters are mentioned.

m) Page 12, line 23: avoid the second "and".

n) Page 14, line 6: add the meaning of NSAIDs.

o) Page 14, line 21: missing space between "[50]." and "However".

p) Page 15, line 5: missing space between "inhibitors" and the reference "[57]".

2. Acinetobacter species are non-fermenting bacteria. The authors on page 7 mention that they selected lactose fermenting bacteria, which by definition are not Acinetobacter species.

3. The authors divided their strains into 12 major profiles, but they do not explain the criteria followed.

4. The authors must explain why they only choose 34 isolates.

5. If "Evaluation of FQs-antibiotic combinations" and "Evaluation of FQs- non-antibiotic combinations" are subsections of "Evaluation of drug combinations", they should be marked as such.

6. Table 1: the title of the second column is misspelled the words "and" and "before" must change their position; the "D" from "FQDs" should be eliminated.

7. There are missing data in table 2.

8. Figure 5 should be redone, the X-axis has the same name as the Y-axis, and the legend is not clear.

9. The quality of the figures should be improved.

10. The conclusion should not include results.

6. PLOS authors have the option to publish the peer review history of their article (what does this mean?). If published, this will include your full peer review and any attached files.

Reviewer #1: No

Reviewer #2: No

---

## [Author Response · Author response to Decision Letter 0]

17 Jun 2020

Author's Response to Reviewer's comments

On the behalf of all authors, we would like to thank the reviewers for their valuable comments and suggestions that will actually improve and add to this manuscript. Corrections are highlighted in yellow color and all have been included in the revised manuscript 

Reviewer #1: MANUSCRIPT PONE-D-20-04012

Manuscript PONE-D-20-04012 describes the effect of different drug combinations with ciprofloxacin and levofloxacin on the susceptibility of PDR and XDR A. baumannii to the fluoroquinolones CIP and LEV. Propranolol and diclofenac sodium particularly obliterated the resistance of the PDR and XDR strains. No doubt, any study reporting alternative therapies in the treatment of infections caused by A. baumannii, a species of bacteria notorious for causing difficult to treat infections in immune-compromised patients is important judging from the current global scourge antimicrobial resistance. Unfortunately, not minding the importance of the area of focus of this manuscript, it is difficult for me to recommend this paper for publication in PLOS ONE in its present form.

Author response:

Thank you for the reviewer comments and recommendations. As known, the extensive drug-resistant (XDR) pathogens particularly A. baumannii is a major health problem and currently still remains a great obstacle for the physicians for their management and cure since the therapeutics options remain very limited. Therefore, the main objective of this research was to find new alternatives for the control of the respective superbug through testing various drug combinations. All comments elicited by the reviewer have been taken into full consideration and the required corrections have been made and included in the revised manuscript (corrections are highlighted in yellow color in the revised manuscript).

1. The authors did not make judicious use of existing literature to provide adequate background for their study and relate the findings to previous studies in this area.

Author response:

Thank you for the reviewer comment. We updated both introduction and discussion sections to include all relevant literature available and relate the obtained findings to previous literature (introduction; page 4 lines 77-84 and lines 86-98), Discussion; page 24, lines 313-344; p28; lines 414-418, P29, lines 426-429; P30; lines 469-473)

2. They also did not follow the PLOS ONE guidelines for authors in the preparation of their manuscript. For example, PLOS ONE do not use structured abstract as they did in this manuscript. Similarly, the figure titles were not inserted in the manuscript as spelt out in the author's guidelines.

Author response:

We have noticed that both types of abstracts are accepted for PLOS ONE. We have previously published an article in PLOS ONE with graphical abstract. (https://pubmed.ncbi.nlm.nih.gov/30157188/?from_term=aboshanab&from_sort=date&from_pos=9) . however, we changed the abstract style based on the reviewer suggestion. All figure legends have been inserted at the end of the manuscript as recommended (page 40)

3. The manuscript was not line-numbered, making reference to specific sections difficult in the course of revising the manuscript.

Author response:

We have included line numbering in the revised manuscript as recommended.

4. The writing style and language use need considerable improvement as the present presentation is unclear making it difficult to follow the thoughts and ideas that the authors are trying to present in some parts of the manuscript. There are important word omissions, use of the noun resistant in place of the adjectival form of the word, sentences that do not read well etc.

Author response:

Thank you for your valuable comment. The whole manuscript has been revised thoroughly for any grammar or any spelling mistake

5. Authors should please define all abbreviations at first use

Author response:

All abbreviations were firstly mentioned in full sentences and abbreviations were inserted between brackets, and were used consequently in the whole manuscript thereafter.

6. Most important, I am concerned about the number and narrow geographical origin of the isolates of A. baumannii (34) included in the evaluation of drug combination part of the experiment. In my own opinion, this number is not enough to make the broad conclusion reached by the authors in this study about the effects of those combined drug in ameliorating the resistance of the isolates to antimicrobial drugs.

Author response:

We thank the reviewer for his valuable comment. Actually, we have done these combinations on the 34 A. baumannii isolates, the clinical relevant isolates which included, 2 PDR and 32 XDR isolates of the highest resistant pattern (all were resistance to the 19 antibiotics tested except colistin) and genetically not identical as determined by ERIC-PCR. Besides, these antibiotic combinations take a lot of time, effort, and resources and because of limited resources, we carried out these experiments on the respective isolates that isolated from different patients of two largest clinical hospitals in Upper Egypt were not clonal as confirmed by ERIC-PCR. However, we are willing to conduct the obtained promising combinations on a further large number of isolates in the future. This part has been clarified inserted in the manuscript (page27, lines 377-384)

Specific comments for the authors

Abstract

Do not use structured abstract as they did in this manuscript

Author response:

The abstract has been modified into non-graphical one.

Methods section. Method section not detailed enough leaving out important details about the drug combination trial which is supposed to be the main focus of the manuscript. This critical omission made some of the results presented in the results section of the structured abstract orphans and unlinked to the methods section.

Author response:

We thank the reviewer for his valuable comment. This part has been modified according to reviewer recommendation where more details about the method of evaluation of both antibiotic (FIC) and non-antibiotic (MIC decrease factor, MDF) were inserted (page 2, line 41-50). However, due to the word number limitation of the abstract, more details were inserted in the method section of the manuscript (page 8, lines 178-207)

Results section

1. This section contained information that could have been included in the methods section of the abstract.

Author response:

This part has been modified in the non-graphical abstract where both methods and findings were included (page 2, lines, 39-50)

2. What informed the choice of the 34 isolates included in the drug combination trial? It will be important to know how the authors selected the 34 isolates, especially the 32 XDR isolates from among the 68 isolates that were XDR since the authors found out that the isolates are not clonal through ERIC-PCR. Why did they not include any of the MDR isolates in this test? Including as many isolates as possible will certainly have strengthened and validated the findings of this study (see comment no.6 above).

Author response:

We thank the reviewer for his valuable comment. Actually, we have done these combinations on the 34 A. baumannii isolates including, 2 PDR and 32 XDR isolates representing the 12 different resistance profiles obtained. In addition, we have selected the 32 XDR isolates based on their antibiogram profile where we selected those of different patterns of antibiotic sensitivity as previously determined by disc and MIC measurement. We do agree if this experiment was done on the 100 isolates, results will be more reliable and confident. However, each experiment took a lot of time, effort, and resources and was done in triplicate and because of limited resources, we carried out these experiments on respective isolates of different antibiotic sensitivity profiles and not clonal as confirmed by ERIC PCR. However, we are willing to conduct the obtained promising combinations on a further large number of isolates in the future. This part has been clarified inserted in the manuscript (page27, lines 377-384)

3. ........including 2 PDR and 32 XDR showed synergism in 23.5%, 17.65%............ : I could not understand what the authors are trying to pass across to readers here. Do they mean that the drug combination showed synergism in 23.5% etc. of the isolates? Same goes for the combination with CPZ and propranolol.

Author response:

This part has been clarified in the abstract (highlighted in yellow color) page 23 lines 44-47.

4. Does the concentration range after propranolol (0.5-1mg/ml) refer to both CPZ and propranolol?

Author response:

No, we have tested two concentrations of propranolol (0.5 mg/ml and 1 mg/ml). This part has been corrected and clarified in the whole manuscript (page 2, line 48; table 6)

5. The authors concluded that their results will greatly help physicians for (sic) the proper choice of antibiotic combinations for combating A. baumannii but some of the drugs combined such as CPZ and Propranolol are not antibiotics.

Author response:

Actually, we thank the reviewer for this valuable comment. This part has been modified and corrected in the conclusion into "Combinations of CIP or LEV with CPZ, PR or DIC showed synergism in most of the selected PDR and XDR A. baumannii clinical isolates. The obtained results are still promising in vitro study however, these combinations have to be re-evaluated in vivo using appropriate animal models infected by XDR- or PDR- A. baumannii, a life-threatening pathogen. (page 3, line 55-57)

 This a general conclusion. 

Keywords

extensively drug-resistant should be qualified

Author response:

Extensively drug-resistant has been changed into extensive drug-resistant in the whole manuscript.

Introduction

1. The information presented in this section to provide a background upon which this study is built is scanty. Authors should please add more recent published information to this section to provide a more robust background for their study and to set the study within the context of previous studies in the area.

Author response:

Thank you for the reviewer valuable comment. The introduction section has been modified and updated by adding more recent published information and citations to provide a more robust background of other previous studies. (Page 4, lines; 77-84; 86-94; 96-98)

2. Line 3-4: A. baumannii is not a hospital-acquired infection but a bacterium that cause HAI

Author response:

This sentence has been corrected (page 3; line 64-65)

Methods

Bacterial isolates

1. What did the authors do with the incubated plates? This question becomes important in the light of the next section where they reported carrying out phenotypic and genotypic tests to identify the colonies. How many colonies did the authors subject to this identification step? From which of the three different media used did they selected the colonies? How did they prevent the selection of clonal replicates from the three agar media?

Author response:

We thank the reviewer for his valuable comment. Actually, each of the collected clinical specimen was streak-plated on MacConky agar, (Oxoid Limited, England), then only one the suspected colonies of non-lactose fermenter was again steak-plate on blood agar (to ensure no contamination and presence of homologous colonies) and then one colony out of the blood agar (enriched media) was streak-plated on Herellea agar (selective media) for final growth characteristics and colony morphology of A. baumannii. Himedia, India), and incubated at 37°C for 24 hours (reference 20; Gerner-Smidt P, Tjernberg I, Using J. Reliability of phenotypic tests for identification of Acinetobacter species. J Clin microbiol. 1991;29(2):277-82. PMID: 2007635). 

Then one colony out of Herellea was subjected to biochemical tests and finally for genotypic identification via the selection of blaOXA-51 gene using PCR for final confirmation (so one one colony was picked out and run the whole process for final identification) (page 5; lines 115-118).

2. Can the authors supply the name of the positive control used in addition to the ATCC accession number? 

Author response:

A. baumannii ATCC 19606 standard strain was used as a positive control (inserted in the manuscript, page 6; lines 138-139)

3. Antimicrobial susceptibility testing

1. The authors mentioned colistin among the antibiotics used for this test by disc diffusion. This method is not recommended for testing of colistin susceptibility because of the large molecular weight of colistin which hinders its diffusion

Author response:

We thank the reviewer for his valuable comment. We agree with the reviewer that colistin has large molecular weight and this hinders its diffusion to some extend from the disc however, colistin by disc sensitivity was recommended by CLSI 2016 at page 63 under the title “Zone Diameter and Minimal Inhibitory Concentration Interpretive Standards for Pseudomonas aeruginosa”. 

In addition, other studies have used colistin disc diffusion in order to discriminate between XDR and PDR pathogens and these include:

• Falagas ME, Karageorgopoulos DE. Pandrug resistance (PDR), extensive drug resistance (XDR), and multidrug resistance (MDR) among Gram-negative bacilli: the need for international harmonization in terminology. Clin Infect Dis. 2008;46(7):1121-2.

• Piewngam, P., & Kiratisin, P. (2014). Comparative assessment of antimicrobial susceptibility testing for tigecycline and colistin against Acinetobacter baumannii clinical isolates, including multidrug-resistant isolates. International journal of antimicrobial agents, 44(5), 396-401.‏

• Galani, I., Kontopidou, F., Souli, M., Rekatsina, P. D., Koratzanis, E., Deliolanis, J., & Giamarellou, H. (2008). Colistin susceptibility testing by Etest and disk diffusion methods. International journal of antimicrobial agents, 31(5), 434-439.

2. It should not be MIC of XDR and PDR isolates against CIP and LEV, rather it is the MIC of CIP and LEV against the bacterial isolates.

Author response:

Corrected (page 7; lines 155-157)

3. The authors should supply the range of concentrations used for MIC determination

Author response:

The MIC of test concentrations for antimicrobial agents was ranged from 0.125-256 μg/ml. (Inserted in the revised manuscript (page 7; lines, 159-160)

4. The authors mentioned vancomycin among the antibiotics used for MIC determination. Vancomycin is not recommended for use against gram-negative bacteria.

Author response:

We thank the reviewer for his valuable comment. Although vancomycin acts by hindering cell wall synthesis in the Gram-positive pathogens only, however, in previous study vancomycin showed a synergistic effect with colistin against A. baumannii (11) and it was reported that the membrane-permeabilizing properties of colistin could enhance the activity of vancomycin against A. baumannii [11]. Therefore, we included vancomycin here to test its effect in combination with CIP or LEV, and to achieve this had to measure its MIC before measuring the FIC when combined with CIP or LEV. (This part was clarified in the discussion section; page 29 (lines 418-424)

In addition there are some study reports for the effect of vancomycin combination with colistin against MDR pathogens

1. Chen, F., Tang, Y., Zheng, H., Xu, Y., Wang, J., & Wang, C. (2019). Roles of the Conserved Amino Acid Residues in Reduced Human Defensin 5: Cysteine and Arginine Are Indispensable for Its Antibacterial Action and LPS Neutralization. ChemMedChem, 14(15), 1457-1465.‏

2. Gordon, N. C., Png, K., & Wareham, D. W. (2010). Potent synergy and sustained bactericidal activity of a vancomycin-colistin combination versus multidrug-resistant strains of Acinetobacter baumannii. Antimicrobial agents and chemotherapy, 54(12), 5316-5322.‏

Molecular typing of recovered isolates

1. In the abstract section, the authors mentioned that all isolates were not clonal while in this section, it appears that only 34 isolates were subjected to ERIC-PCR. The authors need to clarify this to let us know if they subjected all or 34 isolates to ERIC-PCR

Author response:

We thank the reviewer for his valuable comment. ERIC PCR has been carried out on the selected 34 isolates that were selected ( 2 PDR and 32 XDR with different antibiotic susceptibility patterns). This part was clarified in both the abstract (p2; lines 39-40) and in the Method sections (p7; line 166-167).

2. . Evaluation of drug combinations

1. The authors should please provide a brief summary of how the evaluation of drug combinations was carried out. For example how were the drugs combined? Was the combination carried out in-vitro or in-vivo? These are critical details needed to interpret the results.

Author response:

We thank the reviewer for his valuable comment. A combination of FQs with antibiotics was evaluated by using by Checkerboard method while FQs combination with non-antibiotics was evaluated by MDF method. And both combinations were carried out in-vitro. However, more details have been inserted in the methods and results in order to clarify how the respective drug combinations were evaluated (p8, lines 179-1946 for antibiotic combinations) and (p9; lines 195-207 for non-antibiotic combinations)

2. CPZ was conspicuously missing in the list of drugs evaluated in this section. Interestingly, the main conclusion of the manuscript centered on CPZ and propranolol.

CPZ was inserted and included (p9; line 201)

3. Results

Specimen collection and identification of recovered A. baumannii isolates

1. The authors mentioned isolating 623 non-lactose fermenters and 977 lactose fermenters of which a total of 151 isolates were phenotypically identified Acinetobacter isolates (sic). It is important to clarify if the 151 isolates identified as Acinetobacter were a subset of the non-lactose fermenters or a subset of the lactose fermenters?

Author response:

They were a subset of the non-lactose fermenters. Was clarified this issue in the revised manuscript (p9; lines 216-217).

2. Antimicrobial susceptibility testing

1. The level of resistance to doxycycline was reported as 47% in the abstract but was reported as 57% here. Authors should please clarify which is the correct figure.

Author response:

We are sorry for this typing error. It was corrected in the abstract to 57% (p2; line 38)

2. I am confused that the authors are referring to multiple contamination sources of A. baumannii in Upper Egypt. I am not sure of how this relates to the isolation of the bacteria of the present study from clinical specimens.

Author response:

We thank the reviewer for his valuable comment. Analysis of the resulting susceptibility to 19 antimicrobial agents showed the diversity of the resistance of the isolates. They were divided into 12 major profiles according to the resistance number of antimicrobial agents the resistance extended from 19 to 6 antimicrobial agents. The first profile represents PDR (two isolates) was resistant to 19 antimicrobial agents. The second profile represents some of XDR (32 isolates) was resistant to all tests antimicrobial agents except colistin. Finally, profile number 12 was resistant only to 6 antimicrobial agents. (Fig. 5). (page 10; lines 226-232)

MIC

1. In the methods section, the authors mentioned that they determined the MIC using 7 antimicrobial drugs namely CIP, LEV, ampicillin, ceftriaxone, amikacin, doxycycline, and vancomycin. Curiously in the results section, results for only 2 (CIP and LEV) were given. The authors should please supply results for the remaining 5 antibiotics used in MIC determination.

Author response:

The MIC of the remaining 5 antibiotics has done in order to measure the FIC for evaluating the outcomes of antibiotics combinations. As requested, the MIC values of the 7 antibiotics have been included in table 1 of the revised manuscript.

1. Genotyping

Again, it becomes very important for the authors to clarify how many isolates were genotyped in the methods section. The authors made references to “obtained isolates” and “% of total isolates”. To my own understanding 100 isolates were obtained in the study, but the information here showed only 34 isolates were genotyped which does not represent all the obtained isolates or the total isolates as alluded to by the authors in this section.

Author response:

We thank the reviewer for his valuable comment. Genotyping was done on the selected 34 isolates (corrected, page 12 line 253) and also in Fig. 6

2. Evaluation of drug combinations

1. Two antimicrobial profiles selected including...... Selected for what?

Author response:

We thank the reviewer for his valuable comment. This paragraph was rephrased into (Two antimicrobial profiles including, profile 1 ( PDR profile; 2 isolates were resistant to 19 antimicrobial agents) and profile 2 (XDR profile; 32 isolates were resistant to 18 antimicrobial agents) were selected for further study and evaluating the effect of CIP or LEV combination with some selected antibiotics and non-antibiotics (Fig. 5); page 12 (lines 262-265)

3. Table 1- I think before we can conclude on whether there is synergy or antagonism and the extent of the synergy as reported in this table, there is need for baseline references (i.e. the baseline MIC for CIP and LEV before addition of the combination antibiotics) for each isolate. That I did not see in the table. The authors should kindly correct me if I am wrong.

Author response:

Thank you for your comments. Actually, we have included the final outcomes of antibiotic combinations in only one table. However, we included the required data in three tables instead of one table (the baseline MIC for CIP and LEV before the addition of the combination antibiotics) for each isolate are now included in the revised manuscript in tables 1, 2, & 3)

4. The authors should provide legends to define all abbreviations in the tables and figures.

Author response:

All abbreviations have been included in the footnotes of each table and figure.

Table 2- Please refer to comment no2 above. Table 2 is more like what is expected. One wonders why Table 1 is completely different from Table 2 even though they are used to report similar results.

Author response:

Corrected and updated. 

Table 2 became table 7 besides of tables 5 and 6 as baseline data to calculate MIC decrease factor (MDF) after adding MIC of CIP and LEVO before and after combination with non-antibiotics. Table 1 is completely different from Table 2 due to Table 1 calculates ƩFIC a result combination of antimicrobial agents combination according to Hsieh protocol. On the other hand table, 2 calculates MDF as a result combination of antimicrobial agents with non-antimicrobial agents according to Huguet protocol.

5. The results presented by the authors in Table 2 and the manuscript showed that propranolol and diclofenac sodium completely reduced the MIC of CIP and LEV below the susceptibility cut-off value. One is therefore curious as to the reason(s) why the authors decided to lay emphasis on CPZ in the title without any mention of diclofenac sodium.

Author response:

We thank the reviewer for his valuable comment. Our rationale for this is that both CPZ and PR were effective at a low concentration of 200 µg/m for CPZ, 0.5 mg/ml, and 1 mg/ml for PR, respectively) while DIC was only effective only at high concentration (4mg/ ml). However, we have updated this information and included DIC in the title.

6. Labels on the axes of some of the figures are not expressing any meaning. Both axes of figure 5 carry exactly the same label.

Author response:

We thank the reviewer for his valuable comment. Labels are corrected and updated in the revised manuscript.

Discussion:

1. Again, the discussion of results is not clear and the authors did not use existing literature to place their result in proper perspective.

Author response:

We thank the reviewer for his valuable comment. The discussion section have been thoroughly revised and updated to include all relevant date from previous existing literatures (P24, lines 313-344; page 27, lines 377-384; page 28 (lines 414-429), P29 (lines 426-429), P30 (lines 469-473)

2. If the authors are reporting a total of 100 A. baumannii clinical isolates and 61 were from RTI, UTI and blood, where did the remaining 39 isolates came from?

Author response:

We thank the reviewer for his valuable comment. In our study, a total of 100 A. baumannii MDR clinical isolates of which, 61, 17, 12, 6,2 and 2 isolates were recovered from respiratory tract infection (including Enotreacheal tubes, nasal, sputum and throat, urinary tract infections (including urine and urinary tract catheter), blood, wound, skin, and central venous catheter, respectively. (Inserted in Discussion section P24; lines 299-303)

References

1. The guidelines for authors as regards reference listing was not followed by the authors

 Author response:

We thank the reviewer for his valuable comment. All references have been revised and adjusted according to the journal guidelines

2. The doi provided for reference no 18 is not correct.

Author response:

We thank the reviewer for his valuable comment. It was corrected: Hsieh MH, Chen MY, Victor LY, Chow JW. Synergy assessed by checkerboard a critical analysis. Diagn Microbiol Infect Dis. 1993;16(4):343-9.

https://doi.org/10.1016/0732-8893(93)90087-N

 

 

Reviewer #2: 

1. The manuscript has several typographical errors and English should be reviewed.

Author response:

Thank you for your valuable comment. The whole manuscript has been revised thoroughly for any grammar or any spelling mistake

a) Page 4, line 2: the coma should be after "progression" not before.

Author response:

Corrected (P4, line 86)

b) Page 4, line 16: delete the point after "Catheters".

Author response:

deleted (P5, line 109)

c) Page 4, line 24: missing space between "hours" and the reference "[10]".

Author response:

Corrected (P5, line 118)

d) Page 6, line 14: missing space between "guidelines" and the reference "[15]".

Author response:

Corrected page 7 (line 158)

e) Page 7, line 19: delete the point before "(figure 1)".

Author response:

Done (page 9, line 214)

f) Page 10, line 3: delete the semicolon after "CPZ".

Author response:

deleted (page 17, line 273)

g) Page 10, line 3: the second "CIP" should be changed by "LEV".

Corrected (page 17, line 273)

h) Page 10, line 4: omit semicolon after "44.12%".

Author response:

done (page 17 line 274)

i) Page 11, line 11: delete double space in "and mortality".

Author response:

done (page 25 line 324)

j) Page 11, line 12: delete double space in "due to misuse".

Author response:

done (page 25 line 340)

k) Page 11, line 19: misspelling of "originating" it should be "originated"

corrected (page 26 line 348)

l) Page 12, line 1: include a reference to figure 5 when the three major clusters are mentioned.

Author response:

done (page 26, line 356)

m) Page 12, line 23: avoid the second "and".

Author response:

removed (page 27 line 390)

n) Page 14, line 6: add the meaning of NSAIDs.

Author response:

added (page 29 line 443)

o) Page 14, line 21: missing space between "[50]." and "However".

Author response:

done (page 30, line 463)

p) Page 15, line 5: missing space between "inhibitors" and the reference "[57]".

Author response:

done (page 30 line 471)

2. Acinetobacter species are non-fermenting bacteria. The authors on page 7 mention that they selected lactose fermenting bacteria, which by definition are not Acinetobacter species.

Author response:

They were a subset of the non-lactose fermenters. Was clarified this issue in the revised manuscript (p9; lines 216-217).

3. The authors divided their strains into 12 major profiles, but they do not explain the criteria followed.

Author response:

The collected 100 A. baumanni isolates were categorized into 12 resistance profiles based on their resistance patterns to the 19 antibiotics as illustrated in figure 5 and also clarified in the text (page 10, lines 229-7-232) 

4. The authors must explain why they only choose 34 isolates.

Author response:

We thank the reviewer for his valuable comment. The same author reply to reviewer 1 (query 6).

Actually, we have done these combinations on the 34 A. baumannii isolates, the clinical relevant isolates which included, 2 PDR and 32 XDR isolates of the highest resistant pattern (all were resistance to the 19 antibiotics tested except colistin) and genetically not identical as determined by ERIC-PCR. Besides, these antibiotic combinations take a lot of time, effort and resources and because of limited resources, we carried out these experiments on the respective isolates that isolated from different patients of two largest clinical hospitals in the Upper Egypt were not clonal as confirmed by ERIC-PCR. However, we are willing to conduct the obtained promising combinations on further large number of isolates in future. This part has been clarified inserted in the manuscript (page27, lines 377-384)

5. If "Evaluation of FQs-antibiotic combinations" and "Evaluation of FQs- non-antibiotic combinations" are subsections of "Evaluation of drug combinations", they should be marked as such.

Author response:

We thank the reviewer for his valuable comment. This part has been clarified and the two evaluations were separated in both methods (page 8; lines 180-209) and results (page 12, lines 261-281) sections.

6. Table 1: the title of the second column is misspelled the words "and" and "before" must change their position; the "D" from "FQDs" should be eliminated.

Author response:

Table 1 has been changed and updated according to reviewe1 1 request

7. There are missing data in table 2.

Author response:

Table 2 has been changed and updated according to reviewe1 1 request

7. Figure 5 should be redone, the X-axis has the same name as the Y-axis, and the legend is not clear.

Author response:

Thank you for the reviewer comment. Both label axes are modified and corrected in the revised manuscript.

9. The quality of the figures should be improved.

Author response:

We have modified and increased all figure resolutions as possible.

10. The conclusion should not include results.

Author response:

We thank the reviewer for his valuable comment The conclusion section has been modified and updated in the revised manuscript and the result data has been removed however, sometimes we included our findings in order to compare them with the previous research or literature. 

Dear Dr. Monica Cartelle Gestal, Ph.D., Academic Editor, PLOS ONE, 

On behalf of all authors who participated in writing this manuscript, we would like to express our deepest gratitude to the reviewers for their valuable comments that enriched the revising process. All comments and recommendations by reviewers were done and the whole manuscript was revised thoroughly for any typing or grammatical mistakes. On behalf of all co-authors, we appreciate your kelp and effort and we hope that manuscript become suitable to be accepted for publication and we are looking forwards hearing from you soon.

Kind regards 

Corresponding author,

Prof Dr. Khaled Aboshanab (Ph.D.)

---

## [Decision Letter · Decision Letter 1]

9 Jul 2020

PONE-D-20-04012R1

Propranolol, chlorpromazine and diclofenac restore susceptibility of extensive drug-resistant (XDR)-Acinetobacter baumannii to fluoroquinolones

PLOS ONE

Dear Dr. Aboshanab,

Thank you for submitting your manuscript to PLOS ONE. After careful consideration, we feel that it has merit but does not fully meet PLOS ONE’s publication criteria as it currently stands. Therefore, we invite you to submit a revised version of the manuscript that addresses the points raised during the review process.

We look forward to receiving your revised manuscript.

Kind regards,

Monica Cartelle Gestal, PhD

Academic Editor

PLOS ONE

Reviewers' comments:

Reviewer's Responses to Questions

**Comments to the Author**

1. If the authors have adequately addressed your comments raised in a previous round of review and you feel that this manuscript is now acceptable for publication, you may indicate that here to bypass the “Comments to the Author” section, enter your conflict of interest statement in the “Confidential to Editor” section, and submit your "Accept" recommendation.

Reviewer #1: (No Response)

Reviewer #2: (No Response)

2. Is the manuscript technically sound, and do the data support the conclusions?

Reviewer #1: Yes

Reviewer #2: Yes

3. Has the statistical analysis been performed appropriately and rigorously? 

Reviewer #1: N/A

Reviewer #2: Yes

4. Have the authors made all data underlying the findings in their manuscript fully available?

Reviewer #1: Yes

Reviewer #2: Yes

5. Is the manuscript presented in an intelligible fashion and written in standard English?

Reviewer #1: No

Reviewer #2: No

6. Review Comments to the Author

Reviewer #1: MANUSCRIPT PONE-D-20-04012-R1

I commend the efforts put in by the authors to address the issues raised in the original submission of this manuscript. However, I will say by my own assessment that we are only getting close to our goal of generating a publishable manuscript, we are not yet there. The manuscript still needs thorough language editing to make it readable. While the authors have made commendable efforts in addressing the lapses on the science side of the manuscript, I have not really seen much improvement in the writing style and language use aspect of the paper. Largely, the manuscript is still very difficult to follow and readability has not been significantly improved. I appreciate the fact that the authors, like many other authors are not native English speakers, but it is important that they communicate with their audience, it is my considered opinion that the manuscript is not communicating in its present format. My candid suggestion is for the authors is to seek the assistance of a native English speaker with experience in scientific writing, or if they can afford it, a manuscript editing service. Some specific issues that needed attention are enumerated below

Abstract :

Title: Please change extensive to extensively. The authors probably misinterpret my suggestion that they should qualify “extensively drug resistant” in the keyword, hence they went ahead to change the phrase throughout the manuscript to “extensive”. What I meant is that they should add what is extensively resistant to complete the keyword, in this case A. baumannii.

Line 27: change “of” to “caused by”

Line 29-30: “....and evaluating the various combinations to combat ....” evaluating the various combination of what? In addition, authors should please change “evaluating” to “evaluate”

Line 32: Delete “a total of” before 1600

Line 36: Change to “high percentage of bacterial resistance to 19.....”

Line 42: Please include CPZ, PR and DIC among drugs tested in combination with CIP and LEV

Lines 43-44: Please change “have been tested” to “was tested” and delete “including 2 PDR and 32 XDR

Lines 51-53: Delete the sentence starting with in conclusion, it is not necessary.

Line 55: Delete “ The obtained results are still promising in vitro study” and start the sentence with However.......

Line 57: Please delete “a life threatening pathogen”.

Line 58: Delete “extensive drug-resistant” it is more like a repetition of the first keyword XDR-A. baumannii

Selection of isolates for inclusion in the drug combination experiment

I am still not comfortable with the criteria used by the authors in selecting isolates for the drug combination experiment. According to them, they selected the 34 isolates based on drug susceptibility profile. While this is scientific enough, it remains a curiosity how they selected which among the 68 XDR isolates to include since according to them (Lines 230-231) all the 68 XDR isolates shared exactly the same susceptibility profile i.e. resistance to all antibiotics except colistin. Again, one wonders why representatives of the MDR strains are not included in the study as a 3rd category since the work is about antibiotic resistant A. baumannii?

Introduction

I appreciate that the authors have updates this section of the manuscript with new information. However, I still have concerns about the writing style and presentation which needed considerable improvement.

Line 62: insert the word “bacteria” after aerobic and after common in line 65

Line 157: Delete “against” after isolates

Lines 159-160: Delete “of”

Line 180: Change “a MIC” to “the MIC”

Line 185: did the authors mean to write two instead of tow

Line 193: ....antagonistic when ΣFIC is ˃ than what?

Line 196: The authors should mention the different concentrations of each non-antibiotic drugs tested may be as supplementary information. While they may think this is not very important, it does provide a valuable guide for others who may want to repeat this experiment.

Line 207: Range should be from low to high and not otherwise as presented here.

Lines 214-216: In view of the fact that the target organism in this study is a non-lactose fermeter, I wonder why the authors are mentioning the isolation of lactose fermenting colonies in this section. I will rather they delete all reference to lactose fermenters as it has no relevance in this work.

Lines 223-224: The authors claimed that the lowest resistance was exhibited towards doxycycline at 57% followed by colistin at 5%. 57% could not be lower than 5%

Discussion

The discussion section, much like the whole manuscript still needs considerable improvement in writing style. In addition, much of this section is just a combination of results with previous studies without the authors providing the much needed depth of insight into their results. They also devoted a lot of space to discussion on isolation, antibiotic susceptibility and genotyping of the isolates through ERIC-PCR at the detriment of the real focus of their research which is the evaluation of drug combination in the control of resistant A. baumannii. The resultant effect of this is an overly long discussion section.

Line 321: The authors should please correct the doxycycline susceptibility data to 43% since they earlier mentioned that resistance to doxyxcycline was 57%.

Lines 377-384: This explanation as presently offered does not help make a case for the low number of isolates included in this study. Claiming that the experiment took a lot of time, effort and resources and was carried out in triplicates is at best not tenable. In my own opinion, these are essential ingredients one may not be able to avoid in the business of science. Pointing out that the study provided baseline information upon which a wider study could be based would have been a better alternative.

Line 397: Authors mentioned that their results did not agree previous study in north Egypt. They should please cite that study for reference here. Similarly in Lines 404-405 authors claimed their results agree with previous study in upper Egypt, they should cite that study also for reference.

Lines 414-416: An incomplete sentence

I suppose the values in bracket in Tables 2 and 3 are the baseline MICs of CIP and LEV? If this is so, this should be indicated in the legends to the table.

Figures Legend- I think the authors should check again the journal requirement about figures is that the figure caption and legend be inserted immediately after the paragraph where the figure was first mentioned.

Figure 3. The authors should please correct the value for CT, 95% would not have been that small on the chart.

Reviewer #2: This manuscript has been improved, and has very interesting results, but:

1. The authors should revise again the format of the manuscript. Some spaces are missing, incorrect used punctuation marks, missing words and repetition of others (clearly from copy and paste some sentences), as well as words misspelling.

2. It is not needed to repeat the meaning of the abbreviations along with the text as far they explained the first time.

3. The term “our findings” is not properly used.

4. Line 132. The authors repeat the amount of DNA added to the PCR, this information was provided in line 130.

5. Line 135-136. The sentence implies that they used the UV illuminator before doing the electrophoresis gel.

6. Line 197, 387, 433. “&” is not appropriate for a text, should be changed for “and”, “as well as” or other synonym chosen by the authors.

7. Tables 2 and 3. Authors should add to the table legend the meaning of the numbers between brackets, and use the same names in the table and the legend (CIP-AMP or AMP-CIP).

8. Table 2. There is no A in the table, so there is no point in explaining it in the legend.

9. Table 4. FIC meaning missing.

10. Line 299. I think the authors may want to say bacteremia or blood infection.

11. Lines 397-406. This paragraph was not clearly written, some sentences are incomplete.

12. Lines 426-436. Sentences incomplete. Paragraph unintelligible, clearly a result of copy and paste sentences.

7. PLOS authors have the option to publish the peer review history of their article (what does this mean?). If published, this will include your full peer review and any attached files.

Reviewer #1: No

Reviewer #2: No

---

## [Author Response · Author response to Decision Letter 1]

14 Jul 2020

Author's Response to Reviewer's comments

On the behalf of all authors, we would like to thank the reviewers for their valuable comments and suggestions that will improve and add to this manuscript. Corrections are highlighted in yellow color and all have been included in the revised manuscript. 

Abstract :

Title: Please change extensive to extensively. The authors probably misinterpret my suggestion that they should qualify “extensively drug resistant” in the keyword, hence they went ahead to change the phrase throughout the manuscript to “extensive”. What I meant is that they should add what is extensively resistant to complete the keyword, in this case A. baumannii. 

Author response: 

Thank you for the reviewer's comment. We updated the title and updated the key words

 Line 27: change “of” to “caused by” 

Author response:

Thank you for the reviewer's comment. done (page2, line 27) 

Line 29-30: “....and evaluating the various combinations to combat ....” evaluating the various combination of what? In addition, authors should please change “evaluating” to “evaluate”

Author response:

Thank you for your valuable comment. We updated to “ evaluate the various combinations of ciprofloxacin or levofloxacin with antimicrobial agents and non-antimicrobial agents to combat antimicrobial resistance of XDR A. baumannii”.(page 2; lines 30-31)

Line 32: Delete “a total of” before 1600 

Author response:

Thank you for the reviewer's comment. We deleted “a total of”. (page 2, line 33)

Line 36: Change to “high percentage of bacterial resistance to 19.....” 

Author response:

Thank you for your valuable comment. We updated to “a high percentage of bacterial resistance to 19”(page 2, line 37)

Line 42: Please include CPZ, PR and DIC among drugs tested in combination with CIP and LEV

Author response:

Thank you for the reviewer's comment. this part was updated since CPZ, PR, and DIC are among drugs that were tested according to MIC decrease factor (MDF) protocol however the antibiotic combinations were evaluated according to standard protocol of fractional inhibitory concentrations (ƩFICs) (line 43). So we put them in aseprate senctence for more clarifications (page2, lines 47-49)

Lines 43-44: Please change “have been tested” to “was tested” and delete “including 2 PDR and 32 XDR

Author response: 

Thank you for your valuable comment. We updated the sentence (page2, line 43).

Lines 51-53: Delete the sentence starting with in conclusion, it is not necessary.

Author response: 

Thank you for the reviewer's comment. We deleted the sentence.

Line 55: Delete “ The obtained results are still promising in vitro study” and start the sentence with However.......

Author response: 

Thank you for your valuable comment. We updated the sentence (page 3, lines 51-53).

Line 57: Please delete “a life-threatening pathogen”.

Author response: 

Thank you for the reviewer's comment. We deleted the sentence (page 3, line 55).

Line 58: Delete “extensive drug-resistant” it is more like a repetition of the first keyword XDR-A. baumannii

Author response: 

Thank you for the reviewer's comment. We deleted the sentence.

Selection of isolates for inclusion in the drug combination experiment

I am still not comfortable with the criteria used by the authors in selecting isolates for the drug combination experiment. According to them, they selected the 34 isolates based on drug susceptibility profile. While this is scientific enough, it remains a curiosity how they selected which among the 68 XDR isolates to include since according to them (Lines 230-231) all the 68 XDR isolates shared exactly the same susceptibility profile i.e. resistance to all antibiotics except colistin. Again, one wonders why representatives of the MDR strains are not included in the study as a 3rd category since the work is about antibiotic resistant A. baumannii? 

Author response: 

Thank you for your valuable comment. Yes, we selected the 34 isolates based on the drug susceptibility profile. The clinical isolates which were included, (2 PDR and 32 XDR) The selected 32 showed the highest resistant patterns (all were resistant to 19 teste antibiotics except colistin) while other resistant profiles were those which were susceptible to 2 drugs of the 13 drugs tested. So, we selected the isolates of the highest resistance patterns (life-threatening isolates from medical pont of view). In addition , the the selected isolates non-clonal. Future studies will be conducted for testing these combinations on large number of isolates and to evaluated in vivo using appropriate animal model.

Introduction

I appreciate that the authors have updates this section of the manuscript with new information. However, I still have concerns about the writing style and presentation which needed considerable improvement.

Author response:

Thank you for your valuable comment. The whole manuscript has been revised thoroughly for any grammar or any spelling or amy possible typing errors.

Line 62: insert the word “bacteria” after aerobic and after common in line 65

Author response: 

Thank you for the reviewer's comment. added . (page 3, line 59)

Line 157: Delete “against” after isolates

Author response: 

Thank you for the reviewer's comment. deleted . (page 7, line 151)

Lines 159-160: Delete “of”

Author response: 

Thank you for the reviewer's comment. We deleted "of". (page 7, line 156)

Line 180: Change “a MIC” to “the MIC”

Author response: 

Thank you for the reviewer's comment. We changed it. (page 8, line 170)

Line 185: did the authors mean to write two instead of tow

Author response: 

Thank you for the reviewer's comment. Yes, we changed it (page 8, line 175).

Line 193: ....antagonistic when ΣFIC is ˃ than what?

Author response: 

Thank you for the reviewer comment. “when ƩFIC is > 4”, we changed it (page 8, line 183).

Line 196: The authors should mention the different concentrations of each non-antibiotic drugs tested may be as supplementary information. While they may think this is not very important, it does provide a valuable guide for others who may want to repeat this experiment.

Author response: 

We added the different concentrations of non-antimicrobial agents (page 8, lines 186-191).. 

Line 207: Range should be from low to high and not otherwise as presented here.

Author response: corrected (page 9, line 195) 

Lines 214-216: In view of the fact that the target organism in this study is a non-lactose fermeter, I wonder why the authors are mentioning the isolation of lactose fermenting colonies in this section. I will rather they delete all reference to lactose fermenters as it has no relevance in this work.

Author response: 

Thank you for the reviewer's comment. we updated the sentence. (page 9, line 203)

Lines 223-224: The authors claimed that the lowest resistance was exhibited towards doxycycline at 57% followed by colistin at 5%. 57% could not be lower than 5%.

Author response: 

Thank you for the reviewer comment. we updated the sentence (page 9, line 211) 

Discussion

The discussion section, much like the whole manuscript still needs considerable improvement in writing style. In addition, much of this section is just a combination of results with previous studies without the authors providing the much needed depth of insight into their results. They also devoted a lot of space to discussion on isolation, antibiotic susceptibility and genotyping of the isolates through ERIC-PCR at the detriment of the real focus of their research which is the evaluation of drug combination in the control of resistant A. baumannii. The resultant effect of this is an overly long discussion section.

Author response: 

Thank you for your valuable comment. We revised the discussion and we discussed the results according to the available literature however, there were only few studies on fluoroquinolone combination with antimicrobial agents are available. We revised the dicsusion and we did our best to make it more concise about the drug combinations. 

Line 321: The authors should please correct the doxycycline susceptibility data to 43% since they earlier mentioned that resistance to doxyxcycline was 57%.

Author response: 

Thank you for the reviewer's comment. we changed it. (page 24, line 320)

Lines 377-384: This explanation as presently offered does not help make a case for the low number of isolates included in this study. Claiming that the experiment took a lot of time, effort and resources and was carried out in triplicates is at best not tenable. In my own opinion, these are essential ingredients one may not be able to avoid in the business of science. Pointing out that the study provided baseline information upon which a wider study could be based would have been a better alternative.

Author response: 

Thank you for the reviewer's comment. this part of in the discussion have been updated (page 27, lines, 380-384)

Line 397: Authors mentioned that their results did not agree previous study in north Egypt. They should please cite that study for reference here.

Author response: 

Citation has been include.(page 27, line 380)

Similarly in Lines 404-405 authors claimed their results agree with previous study in upper Egypt, they should cite that study also for reference.

Author response: 

Citation has been included (page 27, line 389)

Lines 414-416: An incomplete sentence

Author response: 

Corrected (page 27, line 383-384)

I suppose the values in bracket in Tables 2 and 3 are the baseline MICs of CIP and LEV? If this is so, this should be indicated in the legends to the table. 

Author response: 

The baselines od MICs of CIP and LEV are provided in table 1. In tables, 2 and 3 are the MICs of the two tested antibiotics after their combinations (this was clarified in the legends of tables 2 and 3).

Figures Legend- I think the authors should check again the journal requirement about figures is that the figure caption and legend be inserted immediately after the paragraph where the figure was first mentioned.

Author response: 

The legends of all figures are provided at the end of the manuscript according to the journal requirement.

Figure 3. The authors should please correct the value for CT, 95% would not have been that small on the chart.

Author response: 

It was provided as % resistance (5% resistance) = 95% sensetive

 

Reviewer #2: 

This manuscript has been improved, and has very interesting results, but:

1. The authors should revise again the format of the manuscript. Some spaces are missing, incorrect used punctuation marks, missing words and repetition of others (clearly from copy and paste some sentences), as well as words misspelling. 

Author response: 

Thank you for the reviewer's comment. Thank you for your valuable comment. The whole manuscript has been revised thoroughly for any grammar or any spelling or typing errors

.

2. It is not needed to repeat the meaning of the abbreviations along with the text as far they explained the first time.

Author response: 

Thank you for the reviewer's comment. all have been revised and corrected 

3. The term “our findings” is not properly used.

Author response: 

Thank you for the reviewer's comment. We replaced “our findings” by “present study”, and “in this study”. (page 24; line 320, apge 25, line 329)

4. Line 132. The authors repeat the amount of DNA added to the PCR, this information was provided in line 130. 

Author response: 

Thank you for the reviewer's comment. Information was provided in line 130, represent final concentration of DNA (100 ng of genomic DNA,), while Information was provided in line 132, represent the volume of DNA was added to PCR mixture (and 1 µl template DNA). However, we deleted it (containing 100-200 ng of chromosomal DNA) (page 6, line 128). 

5. Line 135-136. The sentence implies that they used the UV illuminator before doing the electrophoresis gel. 

Author response: 

Corrected to " PCR products were analyzed using agarose gel electrophoresis [23]. " (page 6, line 131)

6. Line 197, 387, 433. “&” is not appropriate for a text, should be changed for “and”, “as well as” or other synonym chosen by the authors.

Author response: 

Thank you for your valuable comment. we changed it to “and”in whole manuscript

7. Tables 2 and 3. Authors should add to the table legend the meaning of the numbers between brackets, and use the same names in the table and the legend (CIP-AMP or AMP-CIP).

 Author response: 

There are the MIC of the two antibiotics after their cobmibations. (was clarified and included in the legends of tables 2 and 3)

8. Table 2. There is no A in the table, so there is no point in explaining it in the legend. 

Author response: 

Deleted since we did not have antagonism (deleted)

9. Table 4. FIC meaning missing. 

Author response: 

Thank you for the reviewer's comment. FIC, fractional inhibitory concentration (We added it in the legend of table 4)

10. Line 299. I think the authors may want to say bacteremia or blood infection.

Author response: 

Thank you for your valuable comment. We changed it to a blood infection (page 24, line 298).

11. Lines 397-406. This paragraph was not clearly written, some sentences are incomplete. 

Author response: 

Thank you for your valuable comment. We changed it It has been amended according to reviewer 1 by added references 9 and 10.

12. Lines 426-436. Sentences incomplete. Paragraph unintelligible, clearly a result of copy and paste sentences. 

Author response: 

Revised and corrected.

Dear Dr. Monica Cartelle Gestal, Ph.D., Academic Editor, PLOS ONE, 

On behalf of all authors who participated in writing this manuscript, we would like to express our deepest gratitude to the reviewers for their valuable comments that enriched the revising process. All comments and recommendations by reviewers were done and the whole manuscript was revised thoroughly for any typing or grammatical mistakes. On behalf of all co-authors, we appreciate your help, effort and kind support.

Kind regards 

Corresponding author,

Prof Dr. Khaled Aboshanab (Ph.D.)

---

## [Decision Letter · Decision Letter 2]

12 Aug 2020

Propranolol, chlorpromazine and diclofenac restore susceptibility of extensive drug-resistant (XDR)-Acinetobacter baumannii to fluoroquinolones

PONE-D-20-04012R2

Dear Dr. Aboshanab,

We’re pleased to inform you that your manuscript has been judged scientifically suitable for publication and will be formally accepted for publication once it meets all outstanding technical requirements.

Kind regards,

Monica Cartelle Gestal, PhD

Academic Editor

PLOS ONE

Additional Editor Comments (optional):

Reviewers' comments:

Reviewer's Responses to Questions

**Comments to the Author**

1. If the authors have adequately addressed your comments raised in a previous round of review and you feel that this manuscript is now acceptable for publication, you may indicate that here to bypass the “Comments to the Author” section, enter your conflict of interest statement in the “Confidential to Editor” section, and submit your "Accept" recommendation.

Reviewer #1: (No Response)

Reviewer #2: All comments have been addressed

2. Is the manuscript technically sound, and do the data support the conclusions?

Reviewer #1: Yes

Reviewer #2: (No Response)

3. Has the statistical analysis been performed appropriately and rigorously? 

Reviewer #1: Yes

Reviewer #2: (No Response)

4. Have the authors made all data underlying the findings in their manuscript fully available?

Reviewer #1: Yes

Reviewer #2: (No Response)

5. Is the manuscript presented in an intelligible fashion and written in standard English?

Reviewer #1: Yes

Reviewer #2: (No Response)

6. Review Comments to the Author

Reviewer #1: I commend the efforts of the authors in attending to the comments and suggestions of this reviewer which has greatly improve the quality of this manuscript. Still in the same spirit of improving the quality of the manuscript, authors should just attend to this minor corrections after which the manuscript should be accepted for publication.

Line 40: Authors should please define MDR at first use.

Line 165: please delete “of” after percentage

Line 265: Change resistance to resistant

Line 330: Delete the antimicrobial before susceptibility

Line 350: “comate”? did the authors mean to say Combat?

Figure 5 legend: Change “No of resistance.....” to “No of Resistant....”

Reviewer #2: (No Response)

7. PLOS authors have the option to publish the peer review history of their article (what does this mean?). If published, this will include your full peer review and any attached files.

Reviewer #1: No

Reviewer #2: No

---

## [Editor Report · Acceptance letter]

14 Aug 2020

PONE-D-20-04012R2 

Propranolol, chlorpromazine and diclofenac restore susceptibility of extensively drug-resistant (XDR)-*Acinetobacter baumannii* to fluoroquinolones 

Dear Dr. Aboshanab:

I'm pleased to inform you that your manuscript has been deemed suitable for publication in PLOS ONE. Congratulations! Your manuscript is now with our production department. 

Kind regards, 

on behalf of

Dr. Monica Cartelle Gestal 

Academic Editor

PLOS ONE